# ORCHIDEE-MICT-BIOENERGY: an attempt to represent the production of lignocellulosic crops for bioenergy in a global vegetation model

Wei Li[1], Chao Yue[1], Philippe Ciais[1], Jinfeng Chang[1], Daniel Goll[1], Dan Zhu[1], Shushi Peng[2], Albert Jornet-Puig[1]

[1]Laboratoire des Sciences du Climat et de l'Environnement, LSCE/IPSL, CEA-CNRS-UVSQ, Université Paris-Saclay, 91191 Gif-sur-Yvette, France
[2]Sino-French Institute for Earth System Science, College of Urban and Environmental Sciences, Peking University, Beijing 100871, China

*Correspondence to:* Wei Li (wei.li@lsce.ipsl.fr)

**Abstract.** Bioenergy crop cultivation for lignocellulosic biomass is increasingly important for future climate mitigation, and it is assumed on large scales in Integrated Assessment Models (IAMs) that develop future land use change scenarios consistent with the dual constraint of sufficient food production and deep de-carbonization for low climate warming targets. In most global vegetation models, there is no specific representation of crops producing lignocellulosic biomass, resulting in simulation biases of biomass yields and other carbon outputs, and in turn of future bioenergy production. Here, we introduced four new plant functional types (PFTs) to represent four major lignocellulosic bioenergy crops, eucalypt, poplar and willow, *Miscanthu*s, and switchgrass, in the global process-based vegetation model, ORCHIDEE. New parameterizations of photosynthesis, carbon allocation and phenology are proposed based on a compilation of field measurements. A specific harvest module is further added to the model to simulate the rotation of bioenergy tree PFTs based on their age dynamics. The resulting ORCHIDEE-MICT-BIOENERGY model is applied at 296 locations where field measurements of harvested biomass are available for different bioenergy crops. The new model can generally reproduce the global bioenergy crop yield observations. Biases of the model results related to grid-based simulations versus the point-scale measurements and the lack of fertilization and fertilization management practices in the model are discussed. This study sheds light on the importance of properly representing bioenergy crops for simulating their yields. The parameterizations of bioenergy crops presented here are generic enough to be applicable in other global vegetation models.

# 1 Introduction

Biomass-derived fuels serve as an alternative energy source to substitute fossil fuel and are used by many countries to meet renewable energy and climate target (Karp and Shield, 2008; Meier et al., 2015; Robertson et al., 2017). Expanding bioenergy crop plantation is considered in future scenarios for energy security and climate change mitigation (Karp and Shield, 2008; Robertson et al., 2017; Smith et al., 2016). For bioenergy production to provide economic and climate benefits, cultivated plants must have a high productivity and a high yield of harvestable biomass (Karp and Shield, 2008; Robertson et al., 2017; Whitaker et al., 2010). The first generation of bioenergy crops usually refers to grain and high-sugar crops like maize and sugarcane (Karp and Shield, 2008). These crops have high nutrient requirements which demand fertilizer additions causing high $N_2O$ emissions to the atmosphere to achieve a high productivity (Melillo et al., 2009; Searchinger et al., 2008). These grain and high-sugar crops are unlikely to be planted in large-scale for the purpose of bioenergy production because of the food demand pressure for fertile land and fertilizer (Alexandratos and Bruinsma, 2012; Gerland et al., 2014; United Nations, 2017). Compared to the first generation, the second generation bioenergy crops, known as lignocellulosic energy crops like giant miscanthus, swithgrass and short-rotation trees, are adapted to a wider range of climatic and soil conditions and require less nitrogen fertilizer (Cadoux et al., 2012; Miguez et al., 2008). Those second generation bioenergy crops have potentials to be deployed on marginal lands to avoid direct and indirect land use change (LUC) carbon emissions and damage of ecosystem services (Robertson et al., 2017). They also appear to have less greenhouse gas (GHG) emissions and higher energy efficiency than the first generation bioenergy crops (Whitaker et al., 2010).

Bioenergy with carbon capture and storage (BECCS) is the main class of future negative emission technologies expected to result in net removal of atmospheric $CO_2$ (Smith et al., 2016). BECCS has been extensively assumed in Integrated Assessment Models (IAMs) to develop land-based mitigation scenarios for low warming levels (Fuss et al., 2014; Popp et al., 2014). In most IAMs like IMAGE (Bouwman et al., 2006; Stehfest et al., 2014) and MAgPIE (Klein et al., 2014; Popp et al., 2011), second generation bioenergy crops are used as primary energy carriers (Popp et al., 2014). One output from IAMs is future land use maps based on different environmental, socioeconomic and policy constraints. These land use maps, after being translated into plant functional type (PFT) maps, can be used in grid-based dynamic global vegetation models (DGVMs) to simulate the terrestrial carbon dynamics, biogeochemical (e.g. LUC carbon emissions) and biophysical (e.g. albedo and transpiration changes) effects of land use processes (Brovkin et al., 2013; Wilkenskjeld et al., 2014). Global vegetation models can provide in return to IAMs some valuable information like spatially explicit biomass density, crop yield and water availability (Bonsch et al., 2015, 2016; Stehfest et al., 2014). For example, dedicated bioenergy crop modelling has been implemented in a global vegetation model (LPJml) (Beringer et al., 2011; Heck et al., 2016), to simulate biophysical yields and water availability as input data for MAgPIE (Bonsch et al., 2016).

In most global grid-based vegetation models, there is no dedicated PFTs to represent second-generation bioenergy crops. Instead, these plants are often represented by a generic crop PFT. Biases in simulated biomass production and resulting

carbon and energy balance thus arise when ignoring differences in carbon assimilation and phenology between generic crops and lignocellulosic bioenergy crops. Moreover, lignocellulosic woody bioenergy crops like eucalypt, poplar and willow cannot be properly represented by an herbaceous crop PFT. For example, eucalypt has a high maximum rate of carboxylation ($V_{cmax}$) but relatively low leaf area index (LAI) (Stape et al., 2004; Whitehead and Beadle, 2004). *Miscanthus* on the contrary, has a relatively lower $V_{cmax}$ (Wang et al., 2012; Yan et al., 2015) but a higher LAI (Heaton et al., 2008; Zub and Brancourt-Hulmel, 2010) than eucalypt (Whitehead and Beadle, 2004). Even if both *Miscanthus* and switchgrass are C4 crops, *Miscanthus* can achieve a significantly higher yield than switchgrass because of a higher efficiency of converting intercepted radiation into aboveground biomass than switchgrass (Heaton et al., 2008). The water, nitrogen and light use efficiencies are also higher for *Miscanthus* than for switchgrass, resulting in a higher rate of leaf photosynthesis in the former (Dohleman et al., 2009). All these important differences between lignocellulosic bioenergy crops need to be considered, which calls for having a dedicated new model PFT for each species.

Similarly, the way that harvest is implemented for generic crops in global models (usually removing a fixed fraction of biomass, typically on the order of 50%) cannot be used for bioenergy crops. The harvest index (*HI*, harvested biomass as a fraction of aboveground biomass) is very different for grain crops and herbaceous bioenergy crops. In addition, most vegetation models currently do not account for realistic rotations of ligneous bioenergy plants (e.g. poplar and eucalypt). Modeling the harvest of woody bioenergy crops should be based on rotation practices of typically a few years rather than on assuming annual full harvest like for herbaceous crops. This requires simulating forest age dynamics (Yue et al., 2018) to accurately represent the ligneous biomass harvest.

In this study, we aim to model biomass yields of four major lignocellulosic bioenergy crops in the global dynamic vegetation model ORCHIDEE. We introduce the new bioenergy crop PFTs, adjust the parameters relevant to physiology, phenology and harvest process of bioenergy crops based on observations, and evaluate the simulated biomass yields using a new global dataset of field measurements.

## 2 Model development and parameterization

### 2.1 Model description

The proposed parameterizations of lignocellulosic bioenergy crops are based on an extended version of ORCHIDEE (Krinner et al., 2005) — ORCHIDEE-MICT (Guimberteau et al., 2018) which contains relevant features of gross land use change, wood harvest and forest age classes dynamics (Yue et al., 2018) (Fig. S1). The model simulates energy exchange, water balance and vegetation carbon processes in the ecosystem and is the land surface component of the French Earth System Model (ESM) IPSL-CM (Krinner et al., 2005). The principal processes related to carbon cycling comprise photosynthesis, vegetation carbon allocation, autotrophic and heterotrophic respiration, plant phenology (e.g. leaf onset and

senescence) and litter and soil carbon dynamics (Krinner et al., 2005). ORCHIDEE-MICT further includes high-latitude related processes with new parameterizations of soil carbon vertical discretization, snow processes, and the SPITFIRE fire module (Guimberteau et al., 2018). Importantly, the representation of forest age dynamics in this version (Yue et al., 2018) allows us to simulate wood harvest based on rotation length practices, a prerequisite for simulating the woody yields.

There is another ORCHIDEE version including short rotation coppice poplar plantations (ORCHIDEE-SRC, Fig. S1, De Groote et al., 2015) based on the forest management module (Bellassen et al., 2010), but ORCHIDEE-SRC is more designed for studying specific coppicing processes and is evaluated using only two coppicing sites in Belgium. Although detailed forest management processes are not included in ORCHIDEE-MICT, this version includes explicit gross land use changes, i.e., the rotational transitions from other vegetation types to woody bioenergy crops and periodic clear-cut harvest of forests.
These features are important to study the carbon emissions from bioenergy crop when their areas expand by converting other land use types in future BECCS scenarios. In addition, ORCHIDEE-MICT contains a bookkeeping system to track different forest age classes as separate land cohorts at a sub-grid scale (Yue et al., 2018). This functionality allows simulating the woody harvest based on rotation length tracking individually the carbon stock dynamics of different age classes of forests. In addition to the poplar plantation in Europe in ORCHIDEE-SRC (De Groote et al., 2015), we aimed to include herbaceous
bioenergy crops like *Miscanthus* and switchgrass as well as other woody crops like eucalypt and willow in a more systematic way on the global scale.

Originally, there are 13 plant functional types (PFTs) in ORCHIDEE (Table 1) (Krinner et al., 2005). In order to represent the bioenergy crops, we introduced four new PFTs (Table 1). PFT14 is a tropical tree, representing eucalypt (*Eucalyptus spp.*); PFT15 is a temperate tree representing poplar (*Populus spp.*) and willow (*Salix spp.*); PFT16 and PFT17 are treated as
crops, representing *Miscanthus* and switchgrass (*Panicum spp.*) respectively. The reason for separating *Miscanthus* and switchgrass into two PFTs is that they are significantly different in biomass yields and resource use efficiency (Dohleman et al., 2009; Heaton et al., 2008). The default model equations of the four new bioenergy crop PFTs follow the ones of similar PFTs already defined in the model (Table 1), i.e. tropical broad-leaved evergreen (PFT2) for PFT14, temperate broad-leaved summer-green (PFT6) for PFT15, and C4 crop (PFT13) for PFT16 and PFT17. Some parameters were however adjusted
specifically for their corresponding bioenergy crops based on field experiment or measurement data in **Section 2.3**.

### 2.2 Bioenergy biomass harvest module

The new module represents the periodical harvest of bioenergy crops, consisting of two sub-routines differentiating woody and herbaceous crops. For woody types, harvest is based on simulated forest age classes (see details in Yue et al., 2018). Briefly, each woody PFT is sub-divided into six cohort functional types (CFTs) corresponding to different age classes. The
boundary of age classes is set as being PFT specific and defined based on maximum woody biomass (total of the sapwood and heartwood biomass). When the biomass of a young woody CFT reaches the upper boundary defining its age class, it is moved to the next older CFT, and sequentially until it reaches the oldest CFT (mature). The fractional harvested area of a

woody crop PFT in each grid cell is externally prescribed. Then, the harvest algorithm starts from the second youngest CFT, continues with the next older CFT, and eventually reverts to the youngest CFT until the prescribed harvested area is met. For woody bioenergy crops, we adjusted the fraction of aboveground biomass that is harvested (the harvest index denoted *HI*) and put harvested biomass into a separate bioenergy harvest pool rather than mixing it with the modeled wood product pools existing for forest management harvest (Yue et al., 2018) or with an agricultural product pool for the two crop PFTs (PFT13 and 14, Table 1) as defined by Piao et al., (2009). The non-harvested biomass goes to litter. For herbaceous types, only the *HI* fraction of aboveground biomass is harvested (**Section 2.3.4**) after leaf senescence either at the end of growing season or if climate conditions like drought and low temperature trigger canopy senescence in the model. The remaining part of above- and belowground biomass goes to litter pools. Carbon in the bioenergy harvest pool is released to the atmosphere directly.

## 2.3 Parameterization of bioenergy crops

Most parameters in ORCHIDEE are PFT specific (Krinner et al., 2005). Since we aim to improve the biomass production performance of the four bioenergy crop PFTs, we adjusted parameters controlling carbon assimilation (**Section 2.3.1**), allocation (**Section 2.3.2**), phenology (**Section 2.3.3**) and harvest processes (**Section 2.3.4**) based on observed values at ecosystem or leaf scale (Table 2). The number of observations for each parameter varied due to the availability of data, and the sample may also be biased in terms of different species or climate conditions. For each parameter, we collected observational values by a detailed literature survey and used the observational medians first. We then evaluated the model predictions of biomass yields using yield observations. If there is a bias, we adjusted the parameter value within the observational range to reduce the misfit between predicted and observed yields.

### 2.3.1 Photosynthesis parameters

The photosynthesis process at leaf level for C3 and C4 plants in ORCHIDEE-MICT is based on the extended version (Yin and Struik, 2009) of the Farquhar, von Caemmerer and Berry model (FvCB model) (Farquhar et al., 1980). The related parameters generally follow Yin and Struik (2009) except for the maximum rate of Rubisco activity ($V_{cmax}$) and maximum rate of electron transport under saturated light ($J_{max}$). The setting of $V_{cmax}$ and $J_{max}$ for C3 plants is based on Medlyn et al. (2002) and Kattge and Knorr (2007) in order to account for the acclimation of $V_{cmax}$ and $J_{max}$ to temperature. In ORCHIDEE, $V_{cmax25}$ ($V_{cmax}$ at 25 °C) is prescribed for each PFT, and $J_{max}$ is calculated from the ratio ($r_{JV}$) between $J_{max}$ and $V_{cmax}$:

$$J_{max} = V_{cmax} \times r_{JV} \tag{1}$$

$r_{JV}$ is a function of growth temperature ($T_{growth}$) (Kattge and Knorr, 2007):

$$r_{JV} = a_{rJV} + b_{rJV} \times T_{growth} \tag{2}$$

where $a_{rJV}$ and $b_{rJV}$ is the acclimation parameters derived by fitting data from 36 plant species (Kattge and Knorr, 2007). For C4 plants, no acclimation is considered for $V_{cmax}$ and $J_{max}$, and thus $b_{rJV} = 0$ and $a_{rJV}$ is a fixed value (Table 2).

Because values of $V_{cmax}$ and $J_{max}$ are critical for determining carbon assimilation by bioenergy PFTs, we searched for published experimental data of these parameters for eucalypt, poplar, willow, *Miscanthus* and switchgrass and found 26 observation-based publications with 127 entries for $V_{cmax}$ and 69 entries for $J_{max}$ (Table S1).

Some observations of $V_{cmax}$ and $J_{max}$ were derived at other temperatures (Table S1) than 25 °C, and we thus normalized these two temperature dependent variables to $V_{cmax25}$, $J_{max25}$ ($J_{max}$ at 25 °C) using a modified Arrhenius function from Medlyn et al. (2002) and parameters for C3 and C4 plants from Yin and Struik (2009). The ranges of $V_{cmax25}$, $J_{max25}$ and $r_{JV25}$ ($r_{JV}$ at 25 °C, only for the studies reporting both $V_{cmax}$ and $J_{max}$) are shown in Fig. 1a-c. $V_{cmax25}$ values generally decrease from eucalypt > poplar and willow > *Miscanthus* $\geq$ switchgrass. The interquartile range of $V_{cmax25}$ is large for eucalypt (N = 42) from 75 to 126 μmol m$^{-2}$ s$^{-1}$ and for poplar and willow (N = 30) from 57 to 165 μmol $CO_2$ m$^{-2}$ s$^{-1}$. *Miscanthus* and switchgrass have a relatively smaller interquartile range of $V_{cmax25}$ (17 to 32, N = 38 and 12 to 26, N = 17, respectively). We adjusted the prescribed parameters $V_{cmax25}$ and $a_{rJV}$ (Table 2) for each bioenergy crop PFT using a value close to the median value in the observation dataset (Fig. 1a,c, within a range of 10% of the median values). We also verified that $J_{max25}$ from Equation (1) is also in the range of independent $J_{max25}$ observations (Fig. 1b). Importantly, the observation-based estimates of $V_{cmax25}$ and $J_{max25}$ for *Miscanthus* are significantly larger than for switchgrass (p = 0.02 and 0.09 respectively, Fig. 1a,b). Note that the ranges shown in Fig. 1 could be influenced by the sample size and number of studies.

We also adjusted other parameters including $\theta$ (the convexity factor of the response of rate of electron transport to irradiance), and $\alpha_{(LL)}$ (conversion efficiency of absorbed light into e- transport rate at strictly limiting light) and $g_0$ (residual stomatal conductance when irradiance approaches zero) in the leaf-level photosynthesis equations of ORCHIDEE to match higher productivity based on field measurements or empirical data (Table 2). The detailed effects of these parameters on photosynthesis in the FvCB model can be found in Yin and Struik (2009). In brief, $\theta$ and $\alpha_{(LL)}$ are used in the calculation of $J$ (photosynthesis rate limited by electron transport):

$$J = \frac{\alpha_{(LL)}I + J_{max} - \sqrt{(\alpha_{(LL)}I + J_{max})^2 - 4\theta J_{max}\alpha_{(LL)}I}}{2\theta} \tag{3}$$

Where $I$ is the photon flux density absorbed by leaf photosynthetic pigments. $g_0$ is an intercept related to the estimation of $g_s$ (stomatal conductance):

$$g_s = g_0 + \frac{A + R_d}{c_i - c_{i*}} f_{VPD} \tag{4}$$

Where $A$ is the net photosynthesis rate, $R_d$ is the day respiration, and $C_i$ and $C_{i*}$ are the intercellular $CO_2$ partial pressure and $C_i$-based $CO_2$ compensation point in the absence of $R_d$, respectively. $f_{VPD}$ is factor of the effect of leaf-to-air vapor pressure difference (Yin and Struik, 2009).

Specifically for bioenergy crop PFTs, we increased θ to 0.8 for PFT14 (eucalypt) based on Yin and Struik (2017) and to 0.84 for PFT16 (Miscanthus) based on field measurements from Dohleman and Long (2009). Light use efficiency and productivity are high for bioenergy crops (e.g. see reviews by Forrester, 2013; Heilman et al., 1996; Karp and Shield, 2008; Laurent et al., 2015; Lewandowski et al., 2003; McCalmont et al., 2017; Whitehead and Beadle, 2004; Zub and Brancourt-Hulmel, 2010), and we thus set $α_{(LL)}$ and $g_0$ to the maximum boundary in their ranges from Yin and Struik (2009) to favors high light use efficiency and productivity characteristic of bioenergy cultivars (Table 2).

Morphological plant traits are also of key importance to the canopy-level productivity (Chang et al., 2015). The specific leaf area (*SLA*) in ORCHIDEE is a PFT-specific constant (Krinner et al., 2005). *SLA* for different bioenergy crops from our data compilation (164 entries in Table S1) is shown in Fig. 1d. A factor of 2 is used to convert the *SLA* unit from $m^2\ g^{-1}$ dry matter to $m^2\ g^{-1}$ C. Observation derived *SLA* for eucalypt is lower than for the other bioenergy crops, and *SLA* for switchgrass is relatively larger. *SLA* is set to the median value of observations for PFT16 (*Miscanthus*) and PFT17 (switchgrass), and close to the 75th percentile value of the data we compiled for PFT14 (eucalypt) and PFT15 (poplar and willow) (Fig. 1d and Table 2).

Another important plant trait for photosynthesis is the leaf orientation, which determines the radiation extinction in the canopy. Although LAI of eucalypts is generally moderate (Anderson, 1981; Stape et al., 2004; Whitehead and Beadle, 2004), leaf angles are nearly close to vertical in mature eucalypts forest (Anderson, 1981; King, 1997), leading to a good distribution of radiation to the lower canopy layers. The light extinction coefficient (*k*) for PFT14 (eucalypt) is therefore set to 0.36 (Table 2) according to the measurement-based estimate by Stape et al. (2004). Similarly, a field study shows the seasonal average *k* ranging from 0.23 to 0.37 for poplars (Ceulemans et al., 1992; Heilman et al., 1996), and a median value of 0.3 was used for PFT15 (Table 2).

### 2.3.2 Carbon allocation parameters

The maximum carbon allocation to leaf biomass is controlled in ORCHIDEE by a pre-defined maximum LAI value (*LAI_max*) beyond which no carbon will be allocated to leaf (Krinner et al., 2005). We adjusted this parameter to match the observed maximum LAI in the field for the four selected bioenergy plants (Table 2). *LAI_max* for PFT14 (eucalypt), PFT15 (poplar and willow), PFT16 (*Miscanthus*) and PFT17 (switchgrass) are set to be 7, 9, 10 and 8, respectively (Ceulemans et al., 1992; Heaton et al., 2008; Heilman et al., 1996; Whitehead and Beadle, 2004; Zub and Brancourt-Hulmel, 2010).

For woody PFTs in ORCHIDEE, the partitioning between aboveground and belowground sapwood biomass is a function of forest age (Krinner et al., 2005):

$$f_{ab,t} = f_{ab,min} + (f_{ab,max} - f_{ab,min}) \times (1 - e^{-t/\tau}) \tag{5}$$

Where $f_{ab,t}$ is the fraction of sapwood allocated to aboveground at age $t$; $f_{ab,min}$ and $f_{ab,max}$ are the minimum and maximum fraction allocated to aboveground (0.2 and 0.8 respectively); and $\tau$ is an empirical parameter. This equation implies that more biomass is allocated to belowground sapwood to develop coarse roots in younger forests. The partition between aboveground and belowground biomass is influenced by resource supply like water and nutrient availability (Litton et al., 2007). For example, belowground carbon allocation in eucalypt is observed to be strongly reduced by irrigation (Barton and Montagu, 2006; Ryan et al., 2010; Stape et al., 2008). Fertilized poplars also showed greater shoot growth than control plots (Coleman et al., 2004). We assumed that bioenergy trees should usually be under intensive management (e.g. irrigation and fertilization) especially in the establishment year (Caslin et al., 2015; Isebrands and Richardson, 2014; Jacobs, 1981). A higher water and nutrient availability then implies a lower investment of biomass on roots for bioenergy trees. In the ORCHIDEE version used here, as there is no specific fertilization or irrigation practice included, the idealized approach chosen to partially account for these managements operations was to reduce $\tau$ in Equation (5) from 5 to 2 years (Table 2) to give a maximum allocation of sapwood biomass to aboveground faster than in the standard version. The difference of these two values is illustrated in Fig. S2. Also because the rotation length for bioenergy trees is usually of several years only (Karp and Shield, 2008), it is reasonable to assume that these plants allocate more biomass aboveground in the first few years. However, trees like poplar and willow can sprout from the remaining stem or root (Isebrands and Richardson, 2014), which is not accounted for in the model. Last, we also adjusted the factor ($\beta$, Table 2) in the exponential function to calculate the soil water stress in ORCHIDEE (Krinner et al., 2005; McMurtrie et al., 1990) to reduce the soil moisture stress on bioenergy trees (Fig. S3).

### 2.3.3 Phenology parameters

An adjustment of parameters related to phenology was performed for the two herbaceous bioenergy PFTs (PFT16 and PFT17, Table 2) to derive the total biomass production for the whole growing season. Lewandowski et al. (2003) and Zub and Brancourt-Hulmel (2010) reviewed growth temperature and growing season length of *Miscanthus* and switchgrass, and found that these two crops have higher cold tolerance and a longer growing season than grasses. Compared to maize, *Miscanthus* has an earlier leaf onset and later leaf fall, and thus its growing season length is 59% longer (Dohleman and Long, 2009). Some *Miscanthus* genotypes need fewer cumulative degree-days for shoot emergence (60 to 118 degree days) and a high frost tolerance (-9 to -6 °C) (Farrell et al., 2006). To account for this frost tolerance and longer growing season, we decreased the growing degree days for leaf onset in the model ($GDD_{onset}$) from 700 (standard value for C4 crop PFT) to 320 degree days (same as the default value for C4 grass PFT in ORCHIDEE) and the critical temperature for leaf senescence ($T_{senescence}$) from 10 to 0 °C for PFT16 and PFT17 (Table 2). Note that we did not set $T_{senescence}$ to be -9 to -6 °C, because frost tolerance was only documented for certain *Miscanthus* genotypes, so we used a conservative value of 0 °C for *Miscanthus* and switchgrass PFTs. In addition, we increased the critical leaf age beyond which leaves enter senescence ($t_{leaf}$) and the

minimum leaf age to allow leaf senescence ($t_{leaf,min}$) to be the same as the default values for C4 grass PFT (PFT11 in Table 1) in ORCHIDEE (Table 2).

### 2.3.4 Biomass harvest

The harvest index (*HI*) determines how much aboveground biomass is harvested. Theoretically, all the aboveground biomass of a lingocellulosic crop can be used for energy production. Some IAMs (e.g. GCAM3.0, Kyle et al., 2011) indeed assume a *HI* of 1 for switchgrass for instance. In practice, harvesting of *Miscanthus* and switchgrass is usually performed in winter and early spring after drying and nutrient recycling through leaf senescence (Lewandowski et al., 2003; Zub and Brancourt-Hulmel, 2010) which leads to a lower biomass at harvest but enhances nutrient conservation. For example, 18%-46% of the nitrogen in *Miscanthus* can be recycled through leaf falling to soil and translocation from shoots to rhizomes (Cadoux et al., 2012). Similar seasonal nitrogen dynamics were also observed for switchgrass (Heaton et al., 2009). In fact, *Miscanthus* is recommended to be harvested between January and March in practice guidelines (DEFRA, 2007). Otherwise, fertilizers have to be applied to amend the nutrient removal from harvest, which is neither cost-effective nor environment-friendly. For bioenergy trees, current harvesting techniques can hardly harvest 100% of aboveground biomass (Caslin et al., 2015; Isebrands and Richardson, 2014; Jacobs, 1981). Following Caslin et al. (2010), Richards et al. (2017), and Zhuang et al. (2013), we used a *HI* of 0.9 (i.e. 90% aboveground biomass is harvested) for all the bioenergy PFTs in ORCHIDEE (Table 2). However, for simulations using future land use maps generated from IAMs, we would recommend setting the *HI* same as in IAMs to be consistent.

The rotation length for eucalypt, poplar and willow varies among different tree types, species, locations and plantation purposes (Caslin et al., 2015; Isebrands and Richardson, 2014; Karp and Shield, 2008; Keoleian and Volk, 2005; Mead et al., 2001). For example, eucalypt and poplar for sawlog and veneer utilization are often on rotations of 8-20 years, depending on regions (Isebrands and Richardson, 2014; Mead et al., 2001). But short rotation coppice bioenergy plantation of poplar and willow have shorter cutting cycles of 3-5 years (Caslin et al., 2015; Isebrands and Richardson, 2014; Karp and Shield, 2008; Keoleian and Volk, 2005). A rotation length of 8 years was used in LPJml model for bioenergy trees (Beringer et al., 2011). In ORCHIDEE, the rotation length for bioenergy tree PFTs is associated with the setting of age classes (see **Section 2.2**). Namely, harvesting starts from the second youngest age class, thus the age in the second youngest forest age cohort should be set up as same as the rotation length. For idealized simulations presented below, we used a rotation length of 4-6 years based on the harvest age and rotation length in the evaluation dataset (Fig. 2; **Section 3.2**). Here, the harvest age (Fig. 2) represents the age when the biomass of bioenergy trees was harvested or estimated. It is directly reported by the original literature and corresponds to the reported yield. Rotation length (Fig. 2) is the management practice reported in the original literature, and it is the same as harvest age in most studies. In other studies, however, some trees may be harvested earlier or later than the regular rotation length, e.g. for a comparison purpose. In addition, not all literature reported both harvest age and rotation length (see the number of observations in Fig. 2).

## 3 Model evaluation

### 3.1 Evaluation dataset

We used the global bioenergy crop yield dataset from Li et al., (submitted, see **Data availability**) to evaluate the performance of the modified ORCHIDEE-MICT-BIOENERGY model. This global dataset was compiled from more than 200 field measurement based studies with five main bioenergy crop types, i.e. eucalypt, poplar, willow, *Miscanthus* and switchgrass (Li et al., submitted). Most of the measurements (>90%) are based experimental trials, especially for *Miscanthus* and switchgrass. About 98% of the compiled observations are reported as the aboveground biomass, and the rest are reported as the total of aboveground and belowground biomass. We thus didn't exclude the observations of the total biomass in the model-observation comparison since their fraction is very low (<2%). The biomass yield in this dataset is compiled in a unit of ton DM (dry matter) ha$^{-1}$ yr$^{-1}$, corresponding to the mean annual biomass yield. For example, if the original literature reported the total harvested biomass of poplar at a certain age, the total biomass amount is divided by age to get the mean annual biomass yield. If the original literature reported the annual harvested biomass of *Miscanthus* for several years, each annual yield is taken as one observation. Note that this dataset does not distinguish the utilization of the plantation (for bioenergy use or for timber / pulpwood). In order to evaluate the simulated biomass yields by ORCHIDEE at half-degree resolution, we calculated the median and range of all observations in each half-degree grid cell containing at least a site of the dataset. Each half-degree grid cell may contain observations from different sites or one site with different species, genotypes, treatments (e.g. different irrigation or fertilization levels). Globally, the number of half-degree grid cells containing observations for PFT14 (eucalypt), PFT15 (poplar and willow), PFT16 (*Miscanthus*) and PFT17 (switchgrass) are 63, 120, 69 and 44 respectively (see maps in **Section 3.5**), giving a total of 296 grid cells (some may be have several crops in common).

### 3.2 Simulation set-up

The set-up for the site-scale simulations in ORCHIDEE-MICT-BIOENERGY is as follows. The model is forced with 30 min time step climatic forcing data, CRU-NCEP v7 (Viovy, 2017) recycling the period of 1990-2000. The CRU-NCEP forcing data is a merged product of CRU TS climate dataset (Harris et al., 2014) and NCEP reanalysis data (Kalnay et al., 1996). Some observation sites have reported mean annual temperature (MAT) and precipitation (MAP), and we verified that these data are consistent with the MAT and MAP from the CRU-NCEP v7 climate forcing data we used (Fig. S4). Thus no bias correction was applied to the CRU-NCEP v7 climate forcing. The soil texture map used in the model is based on the twelve USDA texture classes from Reynolds et al. (2000).

We assumed a homogenous coverage (100%) of one single bioenergy crop PFT in a grid cell covered by the same PFT type as the site observations. We set an annual harvest fraction of 1% of the grid cell each year. The 1% annual harvest fraction is just an artificial value to make sure that there is always forest in second youngest age class available for harvest every year

after a stable rotation is established. We compared the annual harvested biomass in bioenergy harvest pool in per area unit, so the harvest area has no influence on our model evaluation. For the bioenergy trees (PFT14 and PFT15), a spin-up of 100 years without harvest was run first to derive biomass evolution in time to define the respective biomass boundaries for age classes in each grid cell (see Yue et al., 2018). The biomass boundaries are grid-cell specific because of the different vegetation growth rates in different grid cells. The six age classes from youngest to oldest are thus set to be corresponding to 0-4, 4-6, 6-10, 10-30, 30-50, and >50 years. We set the second youngest age class that is used in priority for bioenergy harvest (**Section 2.2**) to be 4-6 years (Fig. 2) based on harvest age and rotation length reported by the original publications in the evaluation dataset (Li et al., submitted). After spin-up, the simulations for PFT14 (eucalypt) and PFT15 (poplar and willow) were run with bioenergy harvest process for 50 years because we only harvested the second age class (4-6 yr) and 50 years is long enough to establish a stable rotation. The harvested biomass amount for the last 10 years was used to calculate the median and range of the simulated yields. Note that we artificially harvest 1% of the grid cells each year, and the harvested patches will be planted immediately. After the first 5 years (one rotation length), there is always a fraction reaching a full rotation and ready for harvest. The harvest in the last 10 years thus represents 10 harvest events. We divided the harvested biomass by 5 years (4-6 years in the second youngest age class) to obtain the annual mean yields of PFT14 (eucalypt) and PFT15 (poplar and willow). A carbon-to-dry-matter ratio of 0.5 was used to convert the unit of yields into ton DM ha$^{-1}$ yr$^{-1}$.

For the bioenergy grasses (PFT16 and PFT17) simulations were performed directly (without spin-up) with harvest for 50 years, and similarly, the yields of the last 10 years were used for comparison with site observed values. Note that we aim to assess the performance of simulated biomass yields rather than the state of the carbon pools including litter and soil organic matter, that depend on site history. As litter and soil carbon pools do not influence vegetation productivity in the model, we did not perform a full long spin-up of carbon pools to their equilibrium values.

### 3.3 Simulated bioenergy yields at global level

The simulated bioenergy biomass yields in comparison with field observations for the four bioenergy crops are shown in Fig. 3. The model-observation results generally lie around the 1:1 ratio line (Fig. 3 left panel). Although the regression between modeled and observed medians is not significant with a low $r^2$ value because of the overestimation and underestimation at some sites (Fig. 3 left panel), the difference between the two samples of modelled and observed yields is not significant (t-test, $p>0.17$) and the percent bias (PBIAS, defined as sum of biases divided by sum of observed values, Moriasi et al., 2007) ranges from 2% to 8% for all PFTs, implying that the global distributions of modeled and observed yields are consistent. ORCHIDEE-MICT-BIOENERGY reproduces the frequency distributions of the observed biomass yields across different grid cells well (Fig. 3 right panel). The median observed and simulated biomass yields in all grid cells are 16.0 and 17.5 ton DM ha$^{-1}$ yr$^{-1}$ for PFT14 (eucalypt), 8.4 and 8.3 ton DM ha$^{-1}$ yr$^{-1}$ for PFT15 (poplar and willow), 12.7 and 10.8 ton DM ha$^{-1}$ yr$^{-1}$ for PFT16 (*Miscanthus*), and 8.7 and 9.0 ton DM ha$^{-1}$ yr$^{-1}$ for PFT17 (switchgrass), respectively. PFT14 (eucalypt) shows

a large spread both in the observed and simulated biomass yields. Some site observation data with high yield (>25 ton DM ha$^{-1}$ yr$^{-1}$) were not reproduced by the model for eucalypt. By contrast, observed and simulated yields for PFT15 (poplar and willow) and PFT17 (switchgrass) concentrate in a relatively narrow range. In addition, the error bars for most sites (67%, 73%, 74% and 64% for PFT14 to PFT17 respectively) reach the 1:1 line (Fig. 3 left panel), implying that at least some observations in these grid cells can be represented by the model.

It should be noted that it is impossible to perfectly reproduce observations in all grid cells, i.e. all dots in Fig. 3 on the 1:1 line, because of uncertainties in the observation dataset, e.g. treatments, genotypes, and local fertilization or irrigation practices as well as in soil characteristics and climate forcing variations prescribed in the model. The error bars of modelled yield (y-axis) come from the range of different harvest years and represents inter annual variability. The error bars of the observations (x-axis) represent the range from different sites, crop species, genotypes and treatments as well as the observation number in each grid cell. It is difficult to systematically synthesize all these factors to give an optimal observed yield in each grid cell. First, different species and genotypes are impossible to be accounted for in a global vegetation model, and thus a further classification of such information would not help the model evaluation. Second, some management practices are difficult to quantify. For example, some studies reported the irrigation as amount per year while some others reported like "irrigating when necessary". The fertilization rates are also difficult to synthesize between different studies because they applied different types of fertilizers, some annually but some in random years. Third, each observation is associated with different managements / treatments, and there is no uniform standard to weight all these different managements. Last, global vegetation models usually run at a half-degree resolution, which may not fully represent the site level climate variations and soil properties.

## 3.4 Biomass-age relationship at site level

A good representation of biomass-age curves for bioenergy trees in the model is crucial to reproduce the yields, especially in the first several years after planting (≤ rotation length). However, most of the observations in the global evaluation dataset were only mean annual yield (Li et al., submitted). This precludes a detailed analysis of biomass dynamics over time for bioenergy trees. We thus selected 22 studies (Table S2) from the evaluation dataset that reported biomass amount of multiple ages (at least two years) and at the same site for eucalypt, poplar or willow. We went through the original articles to derive the biomass-age curves and compared them with the same curves from the model simulations (Fig. 4 and Fig. 5).

There is a good agreement on biomass-age relationship of eucalypt between model and observation in some sites in Australia and China (Site #2, #8-12 in Fig. 4). But the model underestimates the biomass evolution of eucalypt at Site #13 in New Zealand and overestimates it at Site #5-6 in China (Fig. 4). For poplar and willow, there are two long-term (>10 yr) consecutive observation sites in Wisconsin, USA (Site #13 and #15 in Fig. 5), where the model captures the biomass-age relationship well. In some other sites (Site #2, #3, #7, #14 and #17 in Fig. 5), however, the model results only agree with observations for the first few years and then deviate from the observations afterwards. The model generally coarsely

underestimates the biomass of poplar and willow at all ages in the sites in eastern (Site #1 in Fig. 5) and western (Site #5 and #6 in Fig. 5) coastal region of US, in UK (Site #8 to #11 in Fig. 5) and in Sweden (Site #16 in Fig. 5), but overestimates in India (Site #12 in Fig. 5) and at one site in China (Site #18 in Fig. 5).

Possible reasons for the model-observation differences at each site using the information reported in the original studies (see details in Table S2) include the different varieties of species (e.g. genotypes) and management (e.g. fertilization, irrigation or spacing) in the field, which were not explicitly considered in the model. For example, the model overestimates biomass at Site #4 in Fig. 4 because of the large spacing of plantation in the trial at that experimental site (Han et al., 2010), which results in lower biomass yield when converting the unit of ton DM plant$^{-1}$ yr$^{-1}$ to ton DM ha$^{-1}$ yr$^{-1}$. Site #13, #14 and #15 in Fig. 5 are from the same study (Strong and Hansen, 1993), and the model reproduces at Site #13 and #15 but underestimates it at Site #14. It is because the biomass-age curves at Site #13 and #15 are from the average of several genotypes (some have higher yields and some lower), but only one genotype with relatively high yield was planted at Site #14 (Strong and Hansen, 1993), causing an model underestimation at Site #14. In addition, our model seems to systematically underestimate biomass production of willow for the sites in UK (Site #8-11 in Fig. 5). These observed biomass production in UK was based on a range of willow varieties in trial experiments, and the authors (Lindegaard et al., 2011) claimed that the trial experiments generate higher yields than large-scale commercial plantations because of the differences in land quality and practice guidelines (e.g. cutting, harvest index). Despite of some model-observation differences, we emphasized that the modeled biomass-age curves are consistent with observations for most sites within the rotation length.

### 3.5 Maps of differences between simulated and observed yields

The spatial distributions of relative differences between simulated and observed biomass yields are shown in Fig. 6 to Fig. 9 for each PFT. The observations for eucalypt mainly distribute in Brazil, tropical Africa, South Asia and Australia (Fig. 6). ORCHIDEE-MICT-BIOENERGY slightly underestimates biomass yield for PFT14 (eucalypt) in Brazil and overestimates some grid cells in southern China and Australia. Some biomass observations of eucalypts in Australia are obtained from native forests (Li et al., submitted), which may partly explain the overestimation by model.

Poplar and willow are mainly planted in temperate regions like the United States, Europe, and Central and East Asia (Fig. 7). ORCHIDEE-MICT-BIOENERGY underestimates the biomass yields for PFT15 (poplar and willow) in western US but overestimates the yields in eastern US. There is no distinct pattern for the differences between observations and model results in Europe with both underestimation and overestimation across grid cells. But it seems that the simulated biomass yields are lower than observations in Sweden. In Central and East Asia, biomass yields in the inland grid cells are generally underestimated but those in the coastal areas are overestimated.

Most of the observations for *Miscanthus* are from Europe although some trial tests are also available in eastern US and a few in China (Fig. 8). In the US, very slight underestimation of yield was found in the inland areas while overestimation was

more close to the ocean. The model underestimates biomass yields for PFT16 (*Miscanthus*) in UK and South Europe, and slightly overestimates it in other areas in Europe. There are only three grid cells with *Miscanthus* yield observations in China, and they are all largely overestimated in the simulations.

Switchgrass is a native perennial grass in North America (Lewandowski et al., 2003) and thus mainly grows in the US (Fig. 9). There are also very few observations in Europe and East Asia. ORCHIDEE-MICT-BIOENERGY can generally reproduce the biomass yields for PFT17 (switchgrass) in central US but overestimate in eastern US, especially in some grid cells around the Great Lakes. The simulated biomass yields are lower than observations in grid cells in Europe and China but fit well with observations in the grid cell in Japan.

### 3.6 Model-observation difference in different climate bins

The differences between simulated and observed biomass yields for bioenergy crop PFTs in different MAT and MAP intervals are shown in Fig. 10. There is no systematical bias of simulated biomass yields in the climate space except in the climate zones with relatively high MAT and MAP (upper-right grids in Fig. 10b,c) for PFT15 (poplar and willow) to PFT17 (switchgrass). For these PFTs, it seems ORCHIDEE overestimated the yields with MAT > 15 °C and MAP > 1000 mm yr$^{-1}$. The strong underestimation (darker blue color) seems more aligned to the drier regions, especially for poplar and willow (PFT15, Fig. 10b).

The distribution patterns in Fig. 10 also reflect the different climate conditions of growth for these bioenergy crops. Consistent with their physiological characteristics, eucalypts grow in tropical regions (Fig. 6) with MAT > 10 °C and MAP > 500 mm yr$^{-1}$ (Fig. 10). By contrast, poplars and willows grow in temperate regions (Fig. 7) and some under low MAT and MAP (Fig. 10). *Miscanthus*s and switchgrass are usually planted in Europe and US (Fig. 8 and Fig. 9) with moderate MAT and MAP.

We further investigated whether other climate forcing variables in the model impact the model-observation differences using the multiple linear regression method (Table S3) and the regression tree method (Breiman et al., 1984; Pedregosa and Varoquaux, 2011) (Fig. S5). In these two methods, PFT types and nine climate forcing variables (Table S3) were used as independent variables and the relative model-observation difference as dependent variable. The multiple linear regression is non-significant (p = 0.28) with a very low r$^2$ (0.01), suggesting that the variations in the relative model-observation differences is mostly explained by other factors rather than the climate forcing biases used in the model. In the regression tree (Fig. S5), the first discriminator is short-wave radiation but it only split very few samples. Although north wind speed separates a relatively large proportion of samples (Fig. S5), it has little to do with the biomass production in the model. Therefore, results from these two regression methods suggest the model-observation biases are unlikely caused by the model simulation.

## 4 Discussion

### 4.1 Model performance before and after bioenergy crop implementation

In this study, we added four new PFTs to represent the main lignocellulosic bioenergy crops and implemented new parameterizations for each new PFT. As a first step, we evaluated the biomass production from bioenergy crops in ORCHIDEE using a global field measurement dataset. We compared the biomass yields simulated by the new ORCHIDEE-MICT-BIOENERGY with the yields from previous ORCHIDEE version (Fig. 11). In the previous version, bioenergy crops were all taken as herbaceous C4 crop (PFT13), and thus severe overestimation (overestimating 60% on average) occurs for tropical bioenergy trees (i.e. eucalypts, gray squares in Fig. 11a). Although using herbaceous C4 crop generally reproduce the observed biomass yields of poplars and willows (grey squares in Fig. 11b), different carbon dynamics in litter and soil and water and energy balance can be expected.

Using the right tree PFTs for bioenergy trees and right herbaceous PFTs for bioenergy grasses but without new parameterizations also results in significant biases in the simulated yields compared to observations (blue triangles in Fig. 11). Specifically, using the default parameters of previous version is found to largely underestimate biomass yields of all bioenergy trees (blue triangles in Fig. 11a,b). For bioenergy grasses, slight underestimation was found for *Miscanthus* (blue triangles in Fig. 11c) while large overestimation was found for switchgrass (blue triangles in Fig. 11d) with previous default parameters. The large biomass yields of C4 crops in previous ORCHIDEE version (blue triangles in Fig. 11c,d) mainly result from the high $V_{cmax25}$ (Table 2), which is not the reason for the high yields of *Miscanthus* and switchgrass (Fig. 1). We emphasize again that different bioenergy crops achieve high productivities through different pathways based on their plant traits (**Section 2.3**) and it is important to specifically consider these traits by proper parameterizations in the global vegetation models.

### 4.2 Management impacts on parameters

We adjusted some key parameters (e.g. $V_{cmax}$, $J_{max}$ and *SLA*) related to productivity of bioenergy crops based on a collection of field measurements. We only took the medians and the ranges to validate the parameter values in the model but didn't explicitly consider the impacts of management (e.g. fertilization, species) on these parameters, neither in the model nor in the measurements. Here, we summarized some management effects on these parameters for different bioenergy crops based on measurements as follows.

1) *Miscanthus*: Wang et al. (2012) found that biomass yield of Miscanthus increased under nitrogen addition through elevated *SLA*, but fertilization didn't affect $V_{cmax}$, stomatal conductance ($g_s$) or the extinction coefficient ($k$). Yan et al. (2015) measured photosynthesis variables of three Miscanthus species in two experimental fields and found significantly higher $g_s$, $J_{max}$ and $V_{cmax}$ of *Miscanthus lutarioriparius* than *M. sacchariflorus* and *M. sinensis*.

2) Switchgrass: *SLA* differed significantly among nine cultivars of switchgrass but didn't respond significantly to water stress or nitrogen application for individual cultivar (Byrd and May II, 2000). Trócsányi et al. (2009) reported a lower *SLA* of switchgrass from the early harvest than from the late harvest. Hui et al. (2018) investigated leaf physiology of switchgrass under five precipitation treatments and found significantly higher photosynthesis rate and $g_s$ under elevated precipitation but no significant difference under reduced precipitation compared to control plots.

3) Eucalypt: Lin et al. (2013) measured photosynthesis response of six *Eucalyptus* species to temperature and found significantly different $J_{max25}$ and $V_{cmax25}$ among species but non-significant differences in their ratios ($J_{max25}$ / $V_{cmax25}$) and in the temperature response of $J_{max}$ and $V_{cmax}$. With extra nitrogen supply, $J_{max}$ and $V_{cmax}$ of *Eucalyptus grandis* increased significantly, mainly associated with elevated leaf nitrogen content (Grassi et al., 2002). Sharwood et al. (2017) also found that $J_{max}$ and $V_{cmax}$ of *Eucalyptus globulus* were correlated with leaf nitrogen content and the ratio of $J_{max}$ / $V_{cmax}$ was constant under elevated $CO_2$ or elevated temperature, but *SLA* is influenced by different $CO_2$ and temperature treatments.

4) Poplar and willow: In experimental trials of three *Populus deltoides* clones and two *P. deltoides* × *P. nigra* clones, $J_{max}$ and $V_{cmax}$ of the former species were significantly higher than the latter hybrid despite some clonal variations (Dowell et al., 2009). Wullschleger (1993) summarized the species-specific estimates of $J_{max}$ and $V_{cmax}$, and the five *Populus* species displayed large variations. In a poplar free-air $CO_2$ enrichment (PopFACE) experiment, *P. alba*, *P. nigra* and *P.* × *euramericana* showed significant difference of $g_s$ but non-significant difference of $J_{max}$ and $V_{cmax}$ among species, while the elevated $CO_2$ significantly decreased $J_{max}$ and $V_{cmax}$ but had no influence on $g_s$ species (Bernacchi et al., 2003). *SLA* was also found to differ significantly between *P. deltoides* × *P. nigra* family and *P. deltoides* × *P. trichocarpa* family (Marron et al., 2007). For willows, *SLA* increased significantly under fertilization and irrigation, but the magnitude of response varied among six varieties of *Salix* species (Weih and Rönnberg-Wästjung, 2007). Similarly, the response of *SLA* and $g_s$ to nitrogen fertilization differed among three willow clones, but no significant difference of $V_{cmax}$ was found between fertilization and control plots for all clones (Merilo et al., 2006).

In general, the values of parameters like $V_{cmax}$, $J_{max}$ and *SLA* differ among different species or genotypes within each bioenergy crop type. The parameter responses to management like fertilization and irrigation also show large variations depending on the specific species. Although the effects of management on these parameters seem evident in some cases, a set of quantitative relationships that can be applied in relation to simple management operations in a global vegetation model for large scale and generalized PFT is still lacking. Expanding PFT level to species level in global vegetation models requires substantial computational resource, and more importantly, there may be not enough measured parameters of each species for all the processes implemented in the models. At this stage, therefore, using the medians and ranges across a great number of observations is a more justified and practical way to tune the parameters in the models. But more field measurements and quantitative reviews of relationships between individual parameter and individual management as well as interactions between different parameters and managements are highly needed in future research.

### 4.3 Management impacts on yields

Management like fertilization, irrigation and species plays an important role in the biomass yields. In ORCHIDEE-MICT-BIOENERGY, nutrient limitations and management by irrigation and fertilization are not explicitly implemented. Instead, we used parameter values in the range that favors a higher productivity (**Section 2.3**, Fig. 1) and compared the simulated yields with the median values of all observations regardless the management (Fig. 3). We further categorized the observations into three groups (fertilization, non-fertilization or non-reported) and compared with simulated yields (Fig. S6). There is no systematic bias between simulated yields and yields at fertilized sites for all PFTs (orange dots in Fig. S6). The model seems to overestimate the yields of eucalypt at sites with non-reported information of fertilization (most gray dots above 1:1 line in Fig. S6a, Table S4) and overestimate yields of poplar and willow at sites without fertilization (green dots in Fig. S6b, Table S4). Yields at sites with non-reported fertilization information are underestimated by the model for *Miscanthus* (gray dots in Fig. S6c, Table S4) but overestimated for switchgrass (gray dots in Fig. S6d, Table S4).

We didn't group the observations based on different fertilization rates because there are large variations in the biomass response to fertilization rates. For example, in a quantitative review by Heaton et al. (2004), the relationship between yields of *Miscanthus* and nitrogen application rates were not significant. Cadoux et al. (2012) reviewed 11 studies that measured *Miscanthus* yields under fertilization, and the biomass response to nitrogen fertilization was positive in 6 of the studies but no response in the others. Similarly, some studies showed positive biomass response of poplar to nitrogen fertilization, but others didn't (Kauter et al., 2003). Eucalypt also showed variable response to fertilization while the general response was positive (De Moraes Gonçalves et al., 2004). In quantitative reviews of fertilization effects on yields of switchgrass (Wang et al., 2010) and willow (Fabio and Smart, 2018), the relationship between biomass yields and nitrogen fertilization rates was significantly positive but the coefficient of determination ($r^2$) was very low. In summary, biomass response to fertilization varied largely, and evidence from field measurements is not conclusive. More importantly, the basic soil characteristics should be taken into account in addition to the fertilization rates but unfortunately, we didn't have information of soil nutrient contents nor types, nutrient stoichiometry, rates and timing of applied fertilizers for each site from observations.

We also separated the observations based on irrigation information (irrigation, non-irrigation and non-reported) in comparison with modeled yields (Fig. S7). Both underestimation and overestimation were found for sites with different irrigation management for different PFTs. The yields of eucalypt were underestimated at sites with irrigation (blue dots in Fig. S7a, Table S4) but overestimated at sites with non-reported irrigation information (gray dots in Fig. S7a, Table S4). Compared to fertilization, not many sites reported irrigation information and the quantification of irrigation rates is more difficult. For example, some studies reported irrigation amount per year while some others only reported descriptive information like "soil moisture maintained to field capacity" or "irregular irrigation when necessary".

Comparison between simulated yields and observations for the main species of bioenergy crops is shown in Fig. S8. The model overestimated yields of *Eucalyptus urophylla* × *E. grandis*, *E. globulus* and *E. nitens* (Fig. S8a, Table S5). For poplar

and willow, the model generally overestimated yields of *Populus deltoides* × *P. nigra*, *P. deltoides* but underestimated yields of *P. trichocarpa* and *Salix schwerinii* × *S. viminalis* (Fig. S8b, Table S5). There is underestimation of yields for *Miscanthus* × *giganteus* but overestimation for *Miscanthus sinensis*. In fact, the observed yields of the former are significantly higher than yields of the latter (t-test, $p<0.01$). Only four sites reported yields for Panicum pretense, and they were overestimated by the model (Fig. S8d, Table S5).

## 4.4 Future development

Although the model can generally reproduce the bioenergy crop yields on a global scale, there are still some regional biases of biomass yields for different bioenergy crops. For example, ORCHIDEE-MICT-BIOENERGY underestimates the biomass yields of *Miscanthus* in UK by 43% (Fig. 8) and overestimates the yields of switchgrass in eastern US by 18% (Fig. 9). Thus, for a regional use of modelled results, slight modifications of related parameters would be needed.

In addition to the yields from aboveground biomass, the allocation of belowground biomass also needs to be modified, and the resulting soil carbon stocks need to be evaluated. In the current version, the non-harvested parts of biomass go to the litter pool after each harvest. In reality, however, stumps and coarse roots remain alive in coppicing practices of tree species like eucalypt, poplar and willow, and new shoots grow out of these stumps in the next growing season. Similarly, new shoots grow out of rhizome for perennial grasses like *Miscanthus* in the next growing season after harvest. Carbon in such live biomass compartments does not transfer to the litter or soil and thus does not contribute to soil carbon stocks. It is necessary to correct the model processes in this respect before applying this model to account for the full carbon cycle involving bioenergy plants. Meanwhile, a global observation dataset of belowground biomass and soil organic carbon for bioenergy crops would be desirable to systematically evaluate the model, but does not exists, to the best of our knowledge. For a long-term perspective, the implementation of explicit managements and interactions between bioenergy yields and nutrient limitations are increasingly important to simulate carbon reduction potentials of bioenergy crop deployments.

Beside the biogeochemical processes, it is also critical to further parameterize and evaluate biophysical processes, especially in the coupled simulations of global vegetation models with climate models to calculate the biophysical feed-backs. Field measurements on e.g. leaf traits, heat exchange and transpiration of bioenergy crops extend our knowledge of these biophysical processes and need to be integrated adequately in the global vegetation models.

## 5 Conclusions

Bioenergy crop has been extensively assumed in IAMs and is an important type of future land use. However, most global vegetation models do not have specific representations of these bioenergy crops. It is important to accurately represent the physiology, phenology and carbon allocation of these crops because it fundamentally impacts the hydrology dynamics, energy balance and carbon cycle. Especially for woody bioenergy crops like eucalypts, poplars and willows, not only the

biomass yields but also the seasonal variations and biophysical effects, and carbon turnover are impacted by new parameterizations.

In this study, we demonstrated the importance of proper representative of bioenergy crops in a global vegetation model to reproduce the observation-based biomass yields. We introduced new bioenergy crop PFTs based on their plant characteristics, modified the parameters relevant to productivity based on field measurements and empirical evidence, and added the dedicated harvest process to simulate bioenergy biomass yields. The bioenergy crop simulations in ORCHIDEE-MICT-BIOENERGY generally reproduced the observation-based biomass yields for bioenergy crops at global level. However, it is still difficult to match observations site-by-site due to the uncertainties in the observation dataset and the lack of explicit managements in the model. Evaluations on soil carbon dynamics and biophysical variables are further needed. Our work improves the performance of ORCHIDEE on bioenergy crops modelling, and the parameters used in ORCHIDEE-MICT-BIOENERGY also provide guidance for other vegetation models on incorporating dedicated bioenergy crops.

## 6 Code availability

This model development is based on ORCHIDEE-MICT version (Guimberteau et al., 2018) with gross land use changes and forest age dynamics (Yue et al., 2018). The code availability can be found in these two publications. The newly implemented parameterization can be found in Table 2 in this study. The source code of this version (ORCHIDEE-MICT-BIOENERGY) is available online (http://forge.ipsl.jussieu.fr/orchidee/browser/perso/wei.li/ORCHIDEE_GLUC_BIOENERGY), but its access is restricted to registered users. Request can be sent to the corresponding author for a username and password for code access. ORCHIDEE-MICT is governed by the CeCILL license under French law and abiding by the rules of distribution of free software. One can use, modify and/or redistribute the software under the terms of the CeCILL license as circulated by CEA, CNRS and INRIA at the following URL: http://www.cecill.info.

## 7 Data availability

The compiled $V_{cmax}$ and $J_{max}$ data from observations can be found in supplementary Table S1. The evaluation dataset used in this study i.e. the global yield dataset for major lignocellulosic bioenergy crops based on field measurements, has been submitted to a data description journal and will be freely accessed after the acceptance of the data description paper.

## Acknowledgements

We acknowledge $V_{cmax}$ and $J_{max}$ data compiled by Wullschleger (1993) and in the Biofuel Ecophysiological Traits and Yields Database (BETYdb) (LeBauer et al., 2010) and the related references used in the their datasets are also included in Table S1 (part of the 26 publications).

W.L. and C.Y. was supported by the European Commission-funded project LUC4C (grant No. 603542). W.L., C.Y., D.G. and P.C. acknowledge the European Research Council through Synergy grant ERC-2013-SyG-610028 "IMBALANCE-P".

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

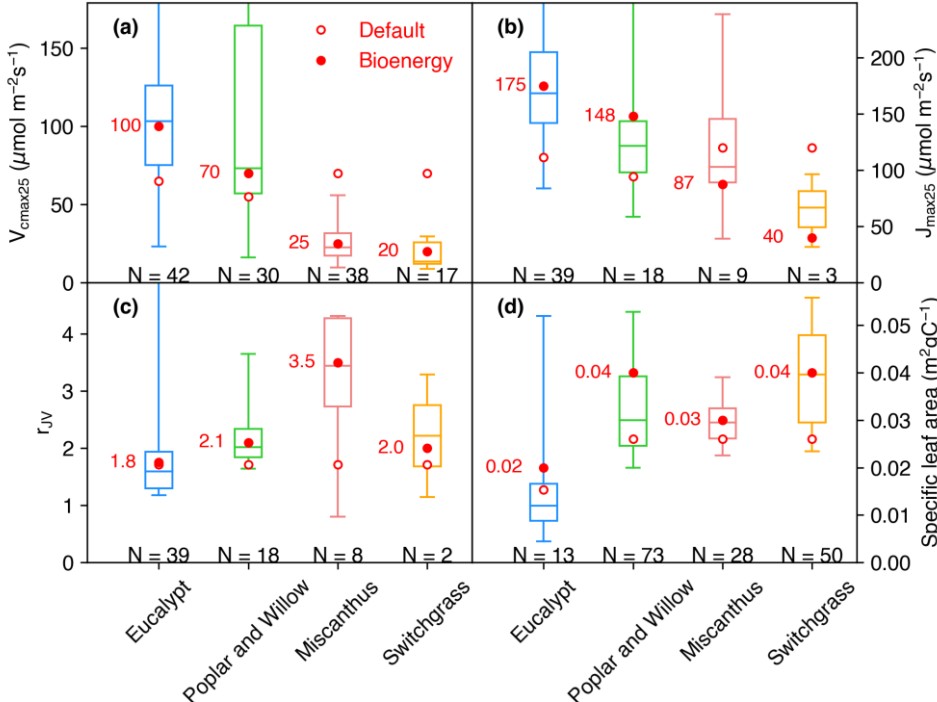

**Figure 1** $V_{cmax25}$ (a), $J_{max25}$ (b), $r_{JV25}$ ($V_{cmax25}/J_{max25}$, c) and specific leaf area (*SLA*, d) collected from measurements. The box plot indicates the interquartile range of measurements. The data size of measurements is shown below the box. The default values (open circles) and adjusted values (filled circles) for bioenergy crops are also shown. Because the model does not prescribe $J_{max25}$ but rather calculates it from $V_{cmax25}$, $J_{max25}$ values for ORCHIDEE shown here (circles in b) are calculated by $V_{cmax25} \times r_{JV}$ (circles in a and b).

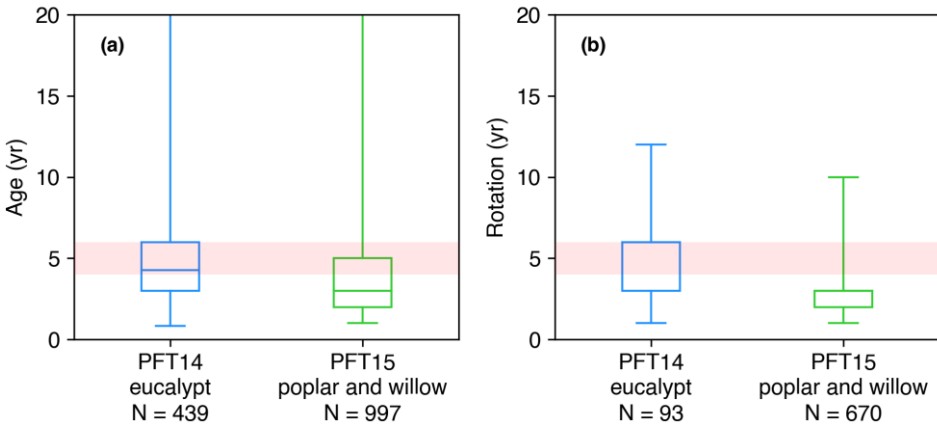

**Figure 2** Harvest age (a) and rotation length (b) in the evaluation dataset. The box plot indicates the interquartile range, and number of observations is also shown. In this study, the harvest age class is set to be 4-6 years (red shade).

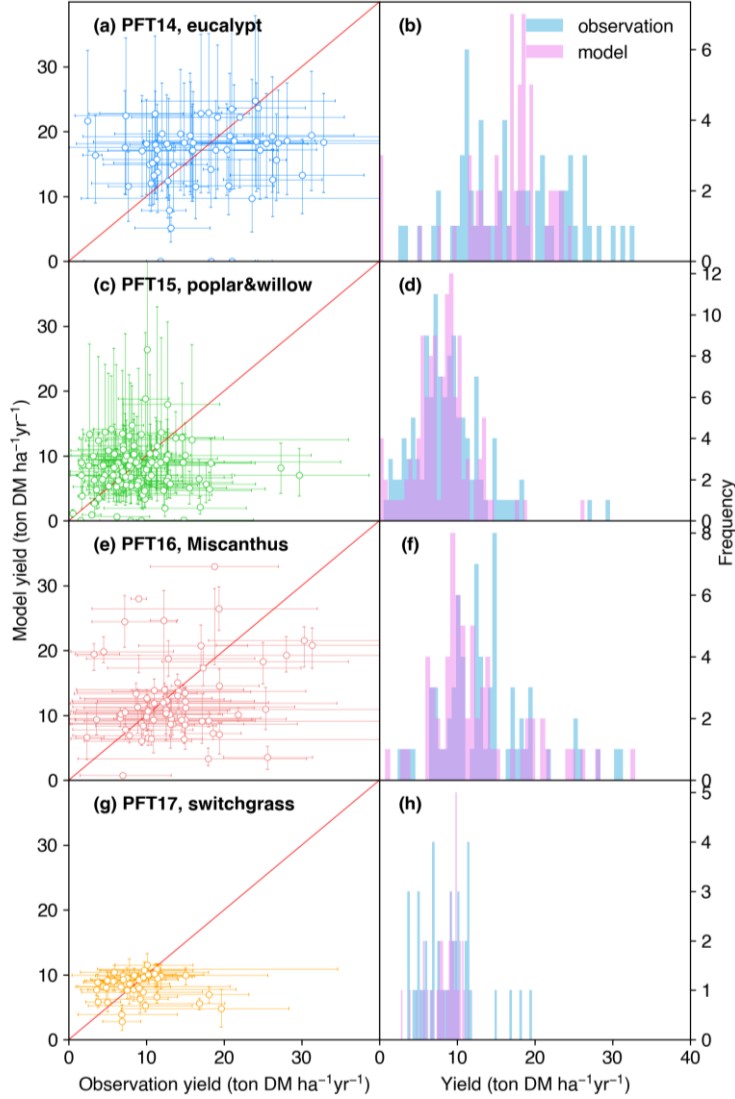

**Figure 3 Biomass yields from the observations and simulated by the ORCHIDEE model. The error bars of observations in the left panel represent the range of different observations in this half degree grid cell caused by different sites, treatments, species and genotypes. The error bars of modelled yields represent the range of different harvest years caused by inter annual variability of climate. PFT 14 is tropical bioenergy tree, eucalypt; PFT15 is temperate bioenergy tree, poplar and willow; PFT16 is C4 bioenergy grass, *Miscanthus*; PFT17 is C4 bioenergy grass, switchgrass. The red line indicates the 1:1 ratio line.**

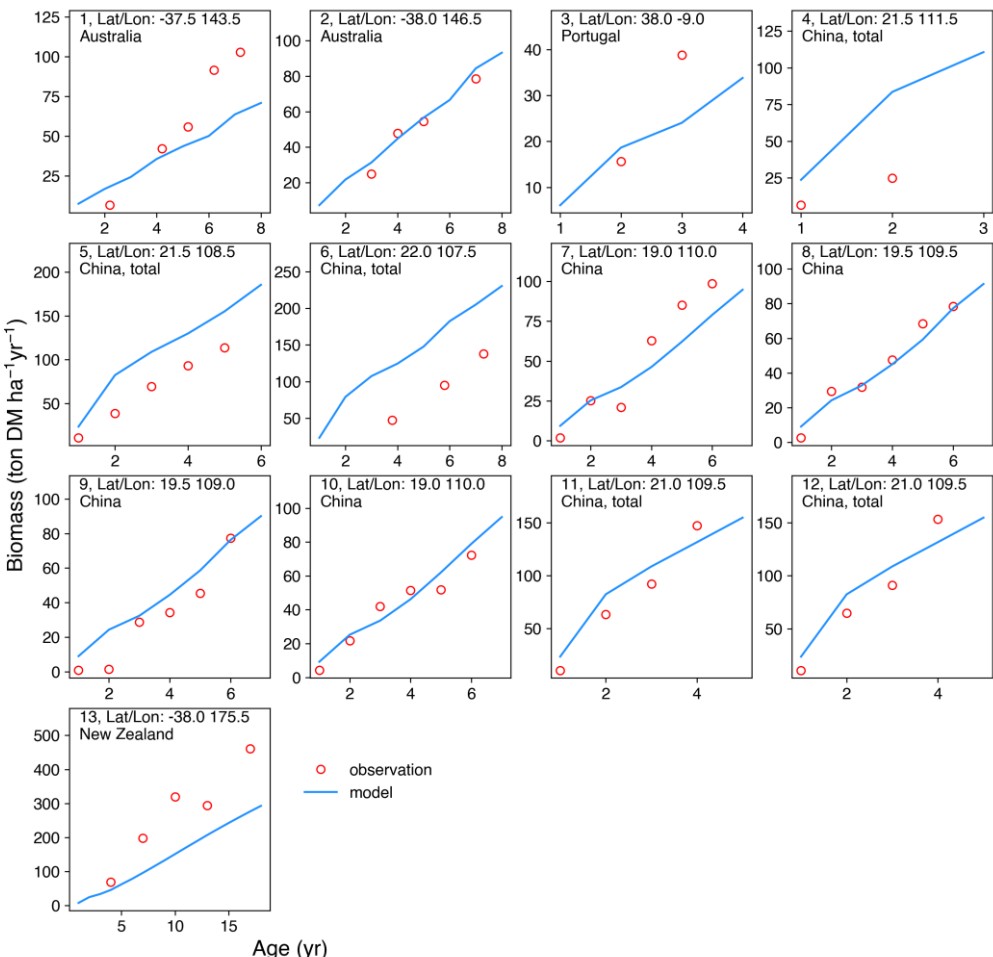

**Figure 4 Biomass-age curves at different sites for PFT14 (tropical bioenergy tree, eucalypt). Site number, coordinates, and country for each site are also shown. Biomass at most sites refers to aboveground biomass, except Site #5, #11 and #12 (labeled "total", i.e. the sum aboveground and belowground biomass; the same total biomass from model is used for these sites). The detailed site information is shown in Table S2.**

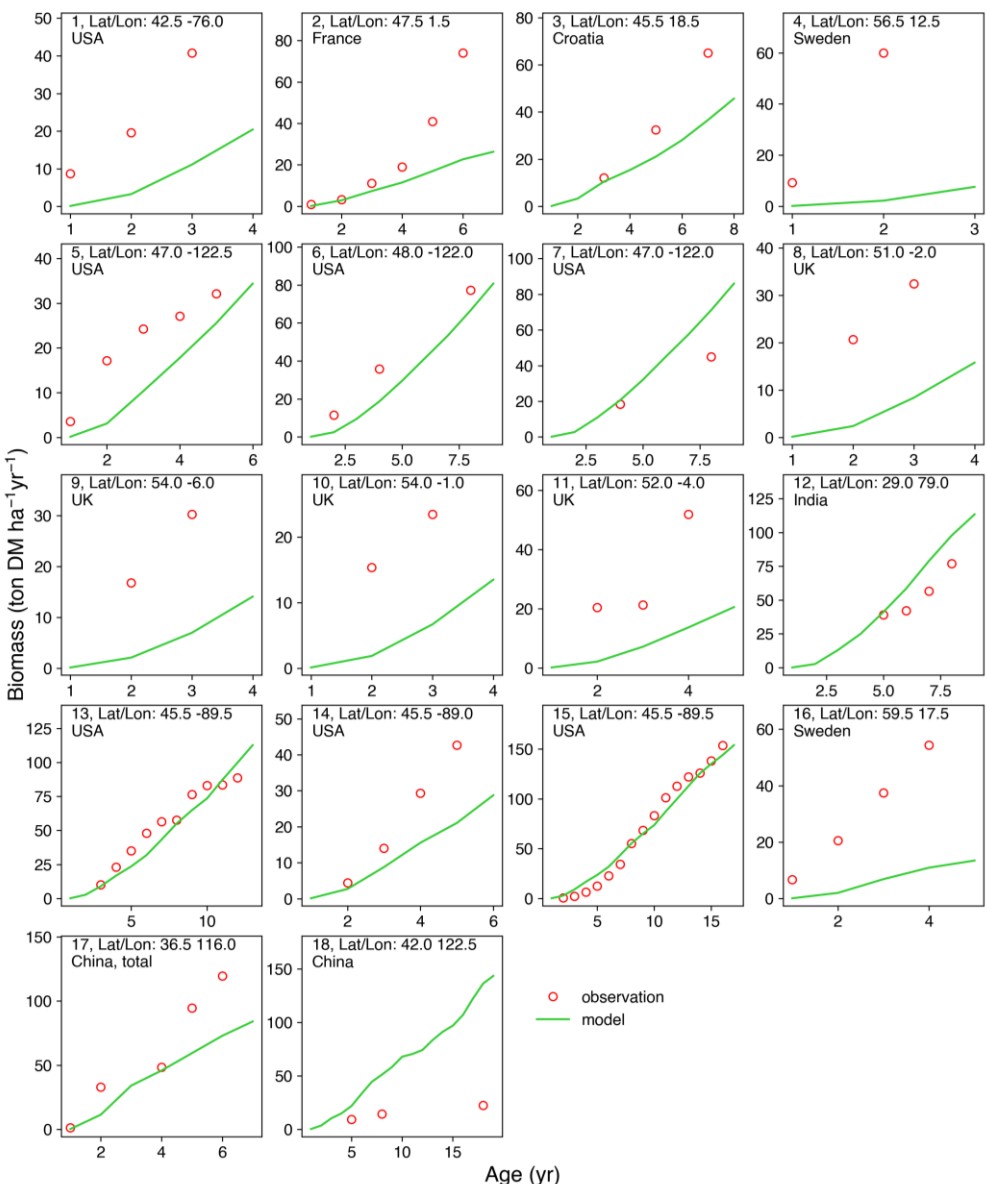

**Figure 5 Biomass-age curves at different sites for PFT15 (temperate bioenergy tree, poplar and willow). Site number, coordinates, and country for each site are also shown. Biomass at most sites refers to aboveground biomass, except Site #17 (labeled "total", i.e. the sum aboveground and belowground biomass; the same total biomass from model is used for this site.). The detailed site information is shown in Table S2.**

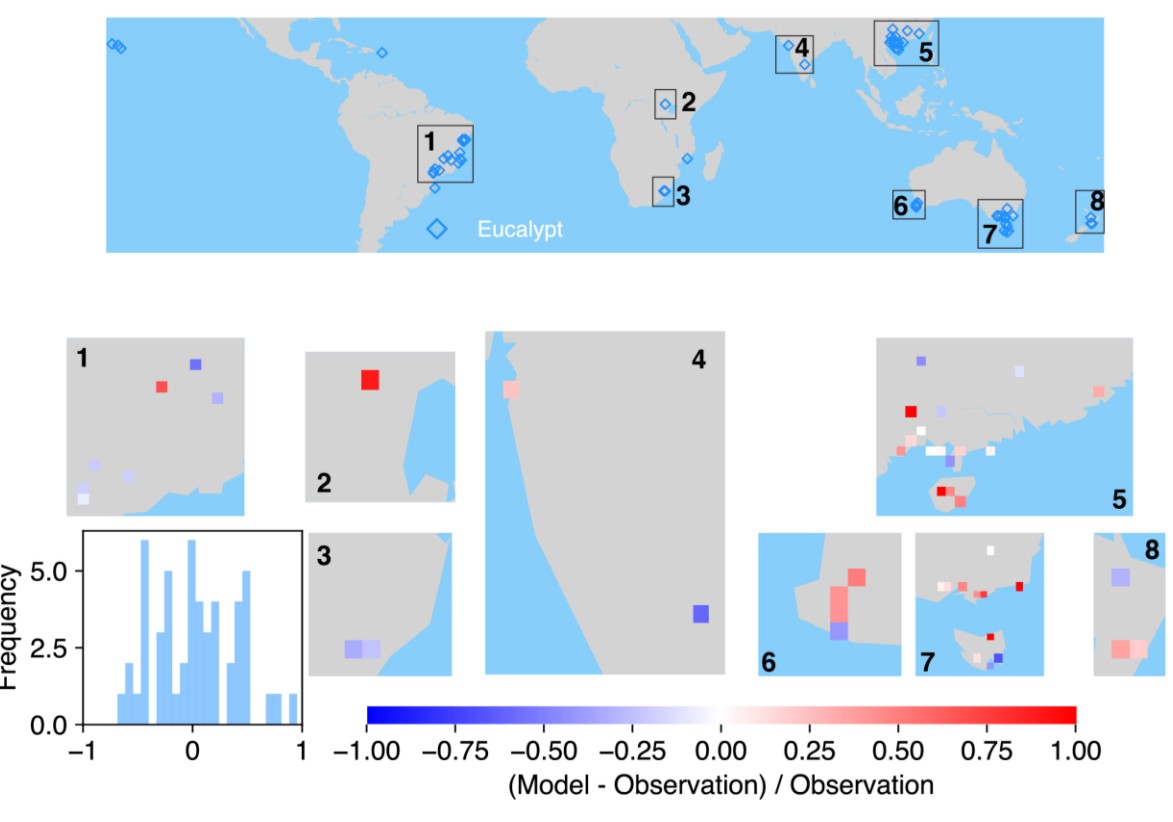

**Figure 6 The map of relative difference between simulated and observed biomass yields for PFT14 (tropical bioenergy tree, eucalypt). The inset plot shows the frequency of the relative difference between model and observation.**

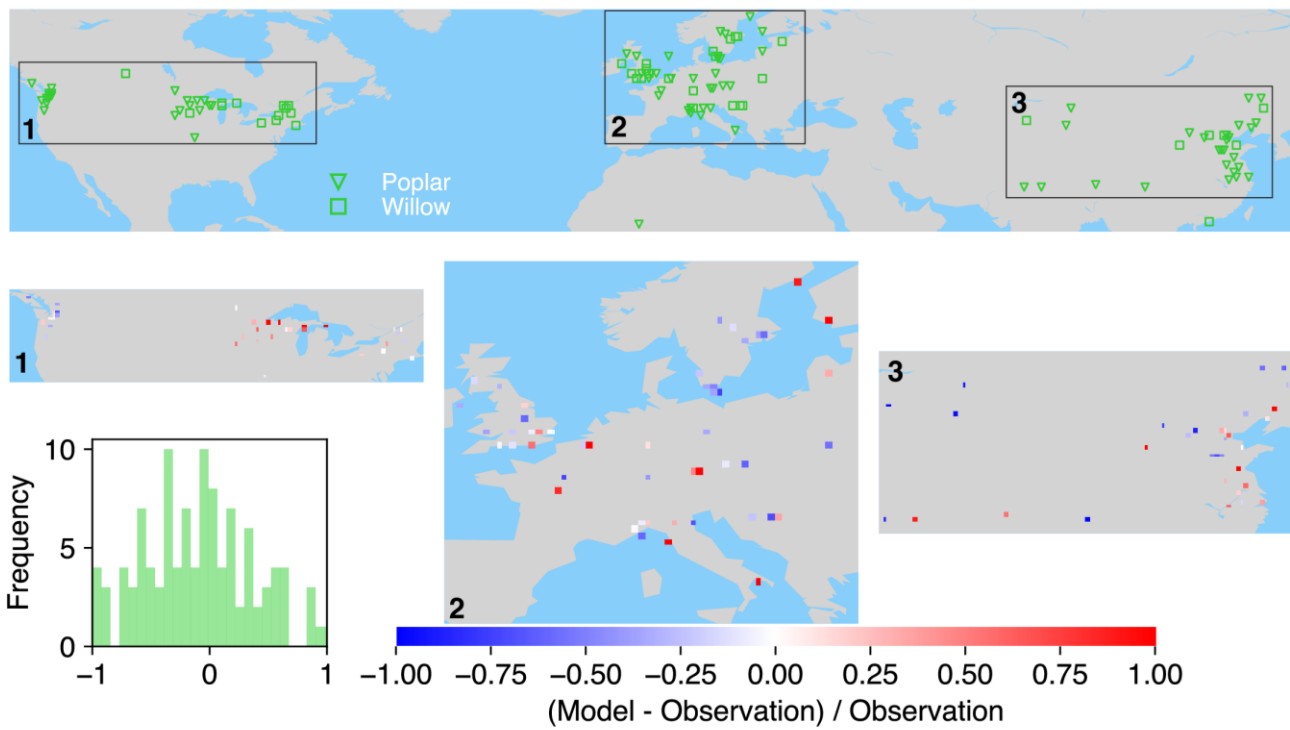

**Figure 7 The map of relative difference between simulated and observed biomass yields for PFT15 (temperate bioenergy tree, poplar and willow). The inset plot shows the frequency of the relative difference between model and observation.**

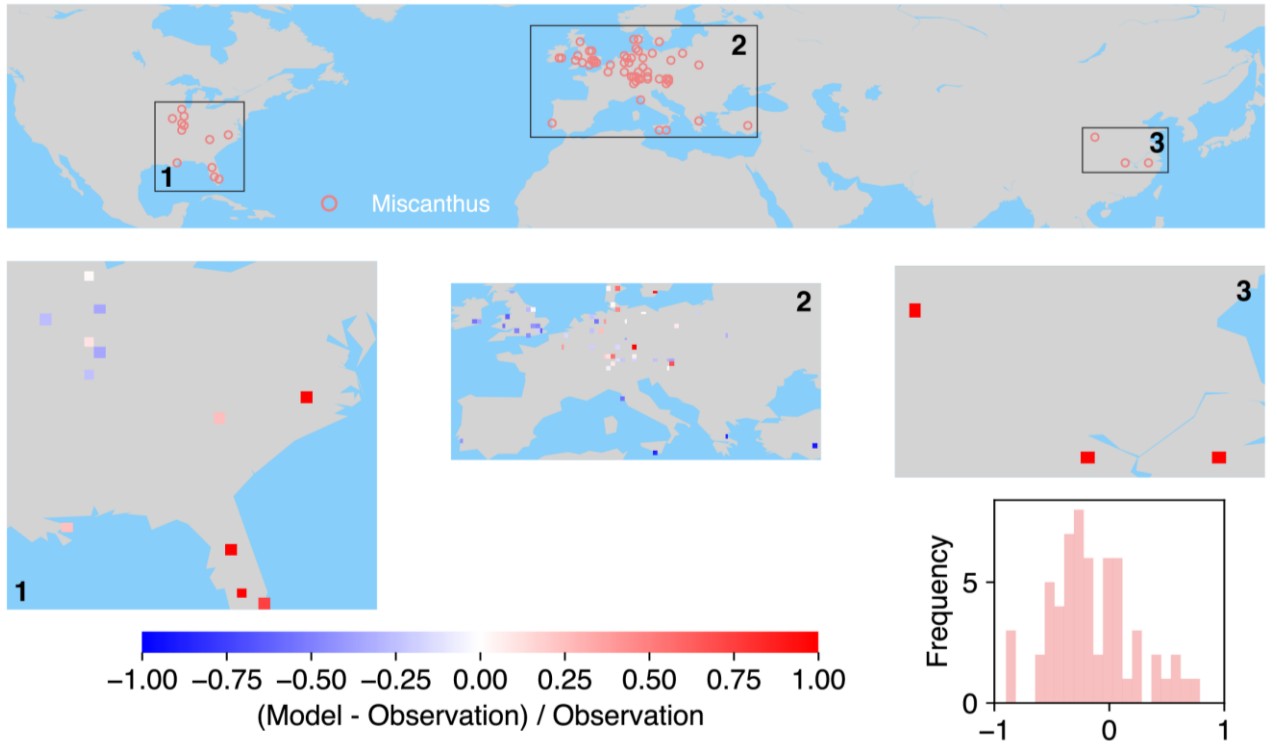

**Figure 8 The map of relative difference between simulated and observed biomass yields for PFT16 (C4 bioenergy grass, *Miscanthus*). The inset plot shows the frequency of the relative difference between model and observation.**

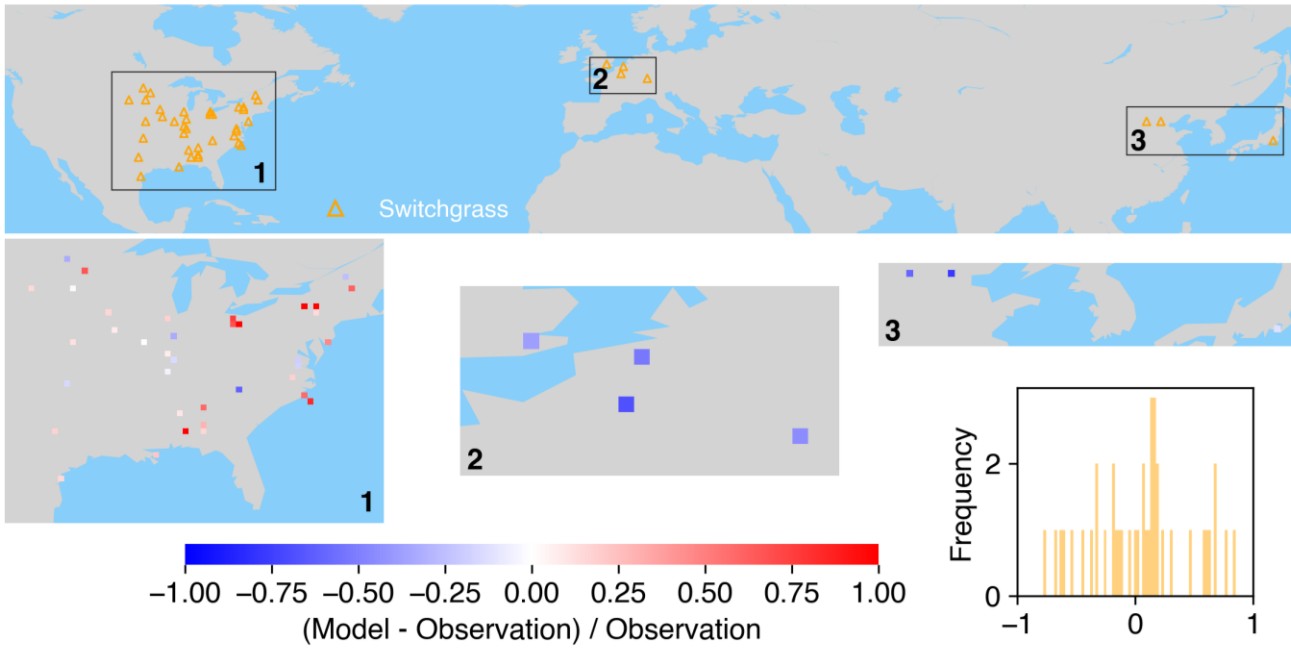

**Figure 9 The map of relative difference between simulated and observed biomass yields for PFT17 (C4 bioenergy grass, switchgrass). The inset plot shows the frequency of the relative difference between model and observation.**

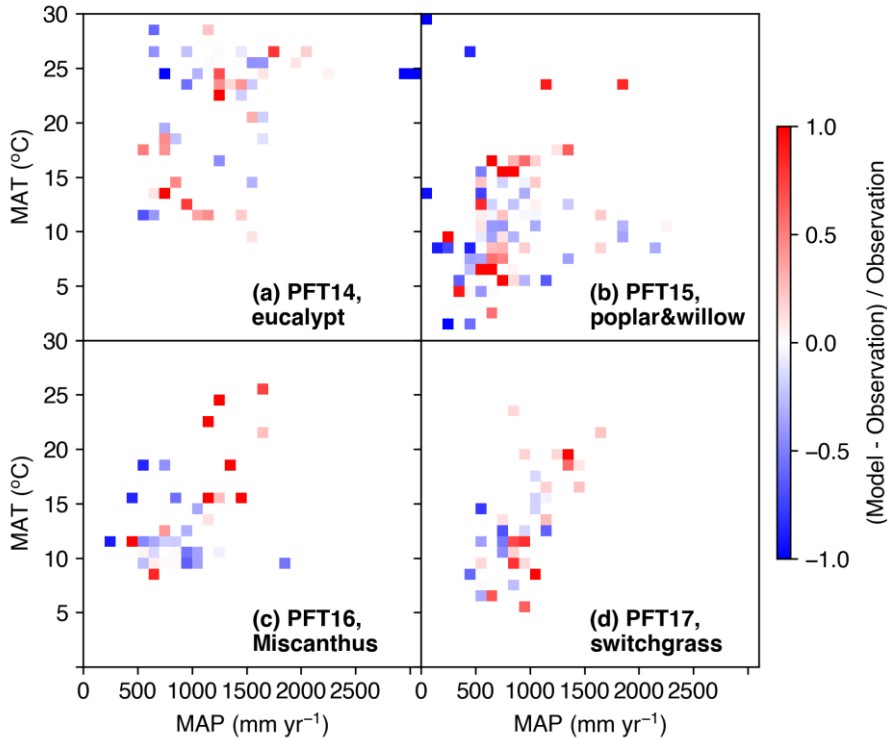

**Figure 10 The relative difference between simulated and observed yield in different MAT and MAP intervals. The median values of model-observation differences of all grid cells in each MAT and MAP intervals are shown. PFT 14 is tropical bioenergy tree, eucalypt; PFT15 is temperate bioenergy tree, poplar and willow; PFT16 is C4 bioenergy grass, *Miscanthus*; PFT17 is C4 bioenergy grass, switchgrass.**

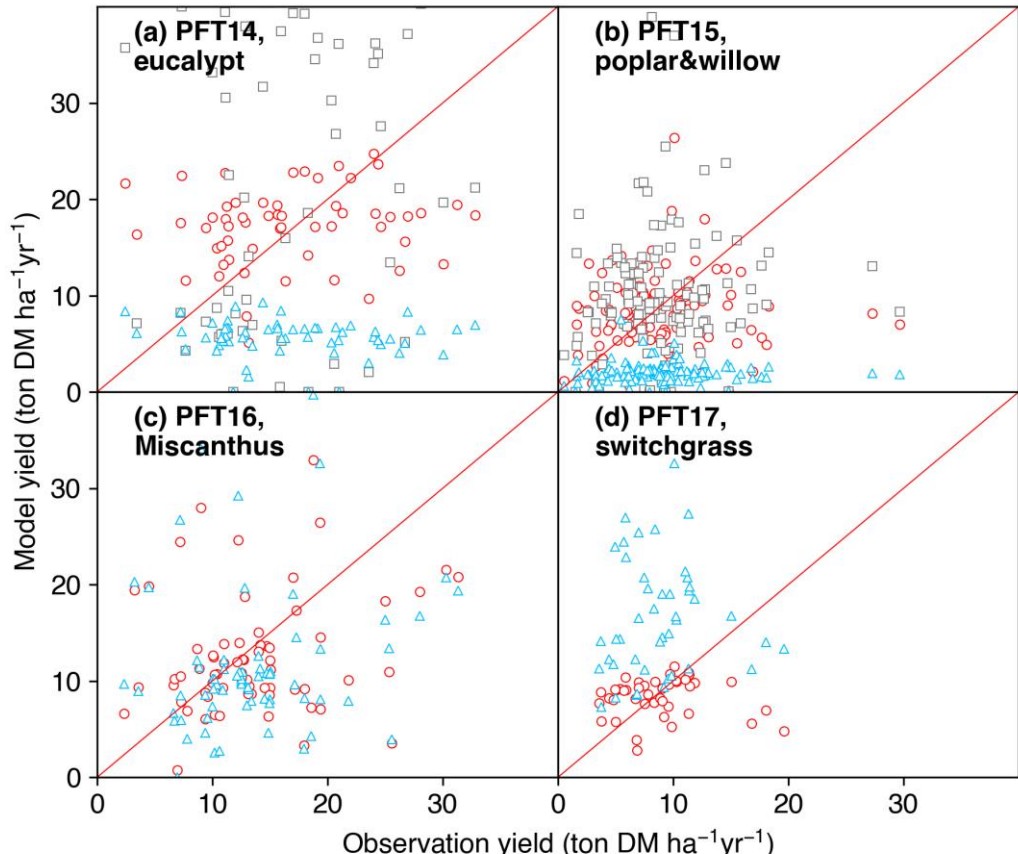

**Figure 11 Comparison of biomass yields simulated by ORCHIDEE-MICT-BIOENERGY and previous versions. Only median values in half-degree grid cells, some containing multiple sites, are shown for both simulated and observed yields. Red circles represent the simulations using specific bioenergy parameterizations (same as Fig. 3). Grey squares represent using the herbaceous crop PFTs of previous ORCHIDEE version for bioenergy trees, i.e. PFT13 for both PFT14 and PFT15 (Table 2). Blue triangles represent the simulations using the right PFTs but the parameters of previous ORCHIDEE version, i.e. parameters of PFT2 (Tropical Broad-leaved Evergreen), PFT6 (Temperate Broad-leaved Summergreen), PFT13 (C4 Crop) and PFT13 (C4 Crop) for PFT14 (eucalypt), PFT15 (poplar and willow), PFT16 (*Miscanthus*) and PFT17 (switchgrass), respectively (Table 2).**

**Table 1. Plant functional types (PFTs) in ORCHIDEE. The newly added bioenergy PFTs (PFT14 to PFT17) use the default setting of the original PFTs (all processes except harvest, see Section 2.2) but with new parameterizations (see Section 2.3).**

| PFT No. | Name |
|---|---|
| 1 | Bare soil |
| 2 | Tropical Broad-leaved Evergreen |
| 3 | Tropical Broad-leaved Raingreen |
| 4 | Temperate Needleleaf Evergreen |
| 5 | Temperate Broad-leaved Evergreen |
| 6 | Temperate Broad-leaved Summergreen |
| 7 | Boreal Needleleaf Evergreen |
| 8 | Boreal Broad-leaved Summergreen |
| 9 | Boreal Needleleaf Summergreen |
| 10 | C3 Grass |
| 11 | C4 Grass |
| 12 | C3 Crop |
| 13 | C4 Crop |
| 14 = 2 | Tropical Bioenergy Tree, representing Eucalypts (*Eucalyptus spp.*) |
| 15 = 6 | Temperate Bioenergy Tree, representing poplar (*Populus spp.*) and willow (*Salix spp.*) |
| 16 = 13 | Bioenergy Grass *Miscanthus* |
| 17 = 13 | Bioenergy Grass Switchgrass (*Panicum spp.*) |

