# Peer review of "ORCHIDEE-MICT-BIOENERGY: an attempt to represent the production of lignocellulosic crops for bioenergy in a global vegetation model"

_Geoscientific Model Development, 2017_

## Referee Comment (RC1) · Anonymous Referee #1 · 6 Mar 2018

The present manuscript is a well written manuscript, which extensively describes the implementation of 4 bioenergy crops of the second generation into the DGVM OR-CHIDEE. The methodology is comprehensively described and the module is validated as good as possible, that makes the manuscript more valuable. Only, it is not clear to me which ORCHIDEE model version is used here. It is not really transparent which version build on which development, as many development papers have recently been published. Could you add something like a development tree for a better understanding? How is the present version related to the version published by De Groote et al.,

2015, which have already introduced a short rotation coppice poplar plantations. Secondly, it was not clear to me how parameters are derived. Some are derived from an observational mean, which is fine, but some I couldn't reproduce where these values come from. Is it a best guess or have you tried to match observational data, but with which method?

Specific comments:

page 4, line 7: Here again, how is the implementation of poplar related of an earlier implementation from De Groote et al., 2015?

page 4, line 26: "The non-harvested biomass goes to litter"-Should that be really the case? In reality you wouldn't plough or something like that to destroy roots respectively non-harvested biomass. Furthermore you would preserve root mass for a faster growth. Especially for woody plantation, growing out of the stump is a coppice management.

page 5, line 15: Doesn't you need the procedure again for the new implemented PFTs?

page 6. line 3: It is not clear to me how the parameters are adjusted and how have you evaluated the adjusted parameters?

Equation 3: Is Jmax = Jmax25? If not, for which equation you need Jmax? Or please do not confuse the reader by defining Jmax25.

page 7, line 28: To allocate only 20 percent to roots seems to me quite small, as the root turnover leads to a higher loss of root biomass. How is root turnover parametrized?

page 7, line 30: I think not to account for growth out of the stump could cause a deceleration of biomass production which is not realistic, but it also causing to high carbon sequestration into the soil.

page 8, line 21: What is the reason for harvesting in winter at lower biomass harvest? Is that really nutrient recycling? I would assume that you can add nutrients in a managed

system.

page 9, line 4: But isn't it less practical to harvest in nearly each age class? The harvester could harm other trees. I would assume that plantations consist of homogeneous age classes and are harvested at a certain age. But maybe I do not understand which practise you assume here.

page 9, line 15: Are there "real" plantations" already or are that more experimental sites?

page9, line 23: "Note that this dataset does not distinguish the utilization .." - But that makes a big difference.

page 10, line 21: It might be better to count the harvest events.

page 10, line 25: But this is of enormous importance if you like to estimate biomass potentials for BECCS. It is essential to balance the harvest and the soil carbon losses and the carbon needed for the establishment of a biomass plantation.

page 13, line 19: But it seems also that the model underestimates yields in dry regions. Blue rectangle tend to be more left sided for PFT15, 16, and 17.

page 14, line 14: "... different carbon dynamics in litter and soil and water and energy balance can be expected." That's why you need to take for the soil carbon balance as well. This is one of the main issue I have on that manuscript.

page 15, line 2: "... global dataset of soil organic carbon for bioenergy crops to our knowledge." At least you should try to represent the carbon cycle right.

De Groote, T., Zona, D., Broeckx, L. S., Verlinden, M. S., Luyssaert, S., Bellassen, V., Vuichard, N., Ceulemans, R., Gobin, A., and Janssens, I. A.: ORCHIDEE-SRC v1.0: an extension of the land surface model ORCHIDEE for simulating short rotation coppice poplar plantations, Geosci. Model Dev., 8, 1461-1471, https://doi.org/10.5194/gmd-8-1461-2015, 2015.

---

## Referee Comment (RC2) · Anonymous Referee #2 · 19 Mar 2018

Overall, this manuscript is a straightforward evaluation of a PFT parameterization in a well-established global biogeochemical model. The authors are adding parameterization of specific plants that are used in lignocellulosic biomass for biofuels. The study is motivated by need to connect a global land biogeochemical model, which typically do not have specific parameterization of biofuel crops, to Integrative Assessment Models that include extensive uses of biofuels in many scenarios for energy development.

I appreciate the authors documenting this model developing through a relatively short publication and that the parameters presented are commonly used across other global

biogeochemical models. This will allow the manuscript serve as a resource for other modeling groups that add these bioenergy crops to their simulations.

My main critique of the manuscript is that it needs more analysis and discussion of causes of the model-data mismatch, specifically the role of management in the parameterization and the observation datasets.

The authors mention that there is considerable variation many of the parameters (e.g., Page 5, line 24). Is that variation related to management? Could there be a parameterization for high intensity management (nutrient additions, irrigation, advanced genetics) and a parameterization for lower intensity management? In general, it would be useful to provide more information about the drivers of variation in the parameters for each species.

The manuscript focuses on a global analysis, rather than comparing directly to individual field studies. By averaging the studies within a grid-cell, there is considerable variation in the observations within a grid-cell (Figure 3). I assume that much of this variation can be attributed to differences in management of the bioenergy crop. For example, there are likely different levels of nutrient fertilization, irrigation, and use of specific genotypes within a grid-cell. I recommend exploring this variation more. Do the simulations compare better to yields from specific types of management? Addressing this question will help set a path for future model development that includes management practices. For example, if the simulations compare better to the nutrient fertilization treatment trials, then including nutrient limitation will potentially help improve the simulations of the biofuel. I realize that the paragraph on page 11, line 9 address this issue but I found paragraph to be weak. Can the studies not be roughly categorized by management intensity? Furthermore, the final sentence "implying the model is able to capture at least some of the observations in these grid cells" does not give much confidence that the new parameterization is actually an improvement.

Also, these is an issue for the editors to provide input on, but the paper leans heavily on

a data paper that is submitted to another unnamed journal. Therefore, a reviewer of this paper is unable to comment on the quality and applicability of the observational dataset. Should this paper be allowed to be published before that data paper is available?

The spatial mapping of the model-bias is useful but it opened the question whether there are spatial differences in management that could explain the spatial variation in the mismatch.

Specific comments

The model evaluation and discussion sections blur together a bit at the edges (section 4.1 seems like a continuation of section 3). I recommend making the separation more clear.

Section 3.3 says that the model-observations results generally lie around the 1:1 ratio line but doesn't provide any statistics on the fit. What is the slope and intercept from the 1:1 fit?

Figure 6. It is hard to see the gridcells in the subboxes. For example, box 2 in Figure 6 has lower points that are impossible to see. Can the subboxes be bigger. I also recommend adding a histogram inset that summarizing the data across grid-cells for all the similar figures (Figure 6-9

Figure 10 stated that there is a 1:1 line that is not present in the figure

Minor comments:

Page 3 Line 25:Change "ORHCIDEE" to "ORCHIDEE"

Page 8, line 22: change 'through leaf falling off' to 'though leaf senescence' Page 9 Line 8: Change "corresponding" to "corresponds".

Page 11 Line 27:Change "after plantation" to "after planting"

Page 12 Line 13:It is unclear what is meant by "because of the large spacing of plantation the trial experiment which results in . . .". Perhaps what was intended was something like: "because of the large spacing of the planting in the trial at that experimental site, which results in . . .".

Page 13 Line 9: Change "US" to "the US

---

## Author Comment (AC1) · 11 May 2018

**Reviewer #1:**

**General Comments:**

**Comment #1**

The present manuscript is a well written manuscript, which extensively describes the implementation of 4 bioenergy crops of the second generation into the DGVM ORCHIDEE. The methodology is comprehensively described and the module is validated as good as possible, that makes the manuscript more valuable.

**Response #1**

We thank the reviewer for the comments and suggestions. Please see the detailed point-by-point responses below.

**Comment #2**

Only, it is not clear to me which ORCHIDEE model version is used here. It is not really transparent which version build on which development, as many development papers have recently been published. Could you add something like a development tree for a better understanding? How is the present version related to the version published by De Groote et al., 2015, which have already introduced a short rotation coppice poplar plantations.

**Response #2**

As we described on **P3L25**: "The proposed parameterizations of lignocellulosic bioenergy crops are based on an extended version of ORCHIDEE (Krinner et al., 2005) — ORCHIDEE-MICT (Guimberteau et al., 2018) which contains relevant features of gross land use change, wood harvest and forest age classes dynamics (Yue et al., 2018)."

As suggested by the reviewer, we will add a figure (reproduced below) to illustrate the origin of ORCHIDEE-BIOENERGY used in this study. The origin of the version by De Groote et al. (*2015*) is also shown in the figure, and the relationship between the two versions are explained below in details. We are aware that there are many other ORCHIDEE development papers, but they are not relevant to the bioenergy version and thus not shown.

**Figure S1** The origin of ORCHIDEE-BIOENERGY version and ORCHIDEE-SRC version.

[Figure]

The version, ORCHIDEE-SRC by De Groote et al. (*2015*), is based on ORCHIDEE-FM, which is an old version for forest management (*Bellassen et al., 2010*). In the forest management module, stand and management characteristics, such as stand density, timing and intensity of thinning, wood removals from stand and post-thinning litter dynamics are simulated (*Bellassen ett al., 2010*). De Groote et al. (2015) further introduced short rotation coppice poplar plantations in that version and evaluated the model using data from two Belgian poplar plantation sites. However, the forest management module is not compatible with ORCHIDEE-MICT, which has the following important extension compared to ORCHIDEE-SRC / ORCHIDEE-FM.

ORCHIDEE-MICT simulates explicitly 1) gross land use change, which is important to simulate the carbon emissions from land use change in future BECCS scenarios, and 2) the age composition dynamics of woody bioenergy crops in relation to their harvest in a grid cell. The explicit spatial

separation of different forest age cohorts allows a proper bookkeeping of different ages of rotation forests and tracking individually their carbon stock dynamics and areal cohorts. In addition, we aimed to introduce additional herbaceous bioenergy crops like *Miscanthus* and switchgrass as well as woody crops like eucalypt, willow and poplar in a more systematic way on the global scale (not only poplar for Europe as in ORCHIDEE-SRC).

We will add sentences to explain the relationship with the version by De Groote et al. (*2015*) on **P3L28**: "There is another ORCHIDEE version including short rotation coppice poplar plantations (ORCHIDEE-SRC, De Groote et al., 20015) based on the forest management module (*Bellassen et al., 2010*), but ORCHIDEE-SRC is more designed for studying specific coppicing processes and is evaluated using only two coppicing sites in Belgium. Although detailed forest management processes are not included in ORCHIDEE-MICT, this version includes explicit gross land use changes, i.e., the rotational transitions from other vegetation types to woody bioenergy crops and periodic clear-cut harvest of forests. These features are important to study the carbon emissions from bioenergy crop when their areas expand by converting other land use types in future BECCS scenarios. In addition, ORCHIDEE-MICT contains a bookkeeping system to track different forest age classes as separate land cohorts at a sub-grid scale (Yue et al., 2018). This functionality allows simulating the woody harvest based on rotation length tracking individually the carbon stock dynamics of different age classes of forests. In addition to the poplar plantation in Europe in ORCHIDEE-SRC (*De Groote et al., 20015*), we aimed to include herbaceous bioenergy crops like *Miscanthus* and switchgrass as well as other woody crops like eucalypt and willow in a more systematic way on the global scale."

**References**

*Bellassen, V., Le Maire, G., Dhôte, J. F., Ciais, P. and Viovy, N.: Modelling forest management within a global vegetation model—Part 1: Model structure and general behaviour, Ecol. Modell., 221(20), 2458–2474, doi:10.1016/J.ECOLMODEL.2010.07.008, 2010.*

*De Groote, T., Zona, D., Broeckx, L. S., Verlinden, M. S., Luyssaert, S., Bellassen, V., Vuichard, N., Ceulemans, R., Gobin, A. and Janssens, I. A.: ORCHIDEE-SRC v1.0: an extension of the land surface model ORCHIDEE for simulating short rotation coppice poplar plantations, Geosci. Model Dev., 8(5), 1461–1471, doi:10.5194/gmd-8-1461-2015, 2015.*

*Guimberteau, M., Zhu, D., Maignan, F., Huang, Y., Yue, C., Dantec-Nédélec, S., Ottlé, C., Jornet-Puig, A., Bastos, A., Laurent, P., Goll, D., Bowring, S., Chang, J., Guenet, B., Tifafi, M., Peng, S., Krinner, G., Ducharne, A., Wang, F., Wang, T., Wang, X., Wang, Y., Yin, Z., Lauerwald, R., Joetzjer, E., Qiu, C., Kim, H. and Ciais, P.: ORCHIDEE-MICT (v8.4.1), a land surface model for the high latitudes: model description and validation, Geosci. Model Dev., 11(1), 121–163, doi:10.5194/gmd-11-121-2018, 2018.*

*Yue, C., Ciais, P., Luyssaert, S., Li, W., McGrath, M. J., Chang, J. and Peng, S.: Representing anthropogenic gross land use change, wood harvest, and forest age dynamics in a global vegetation model ORCHIDEE-MICT v8.4.2, Geosci. Model Dev., 11(1), 409–428, doi:10.5194/gmd-11-409-2018, 2018.*

**Comment #3**

Secondly, it was not clear to me how parameters are derived. Some are derived from an observational mean, which is fine, but some I couldn't reproduce where these values come from. Is it a best guess or have you tried to match observational data, but with which method?

**Response #3**

As described in **Section 2.3**, we did systematic parameterization changes of carbon assimilation, allocation, phenology and harvest based on field measurements / observations. Some parameters have a very limited number of observations while others have substantially more. However, the samples may also be biased in terms of species or climate zones even when a great number of observations exist. So, for each parameter, we first used the observational median and performed model simulations to see if the biomass production matches the observations. If not, we slightly adjusted it again within the observational range to make the modeled values closer to observations.

We will add sentences on **P5L4** to explain this: "The number of observations for each parameter varied due to the availability of data, and the sample may also be biased in terms of different species or climate conditions. For each parameter, we collected observational values by a detailed literature survey and used the observational medians first. We then evaluated the model predictions of biomass yields using yield observations. If there is a bias, we adjusted the parameter value within the observational range to reduce the misfit between predicted and observed yields."

**Specific Comments:**
**Comment #4**

page 4, line 7: Here again, how is the implementation of poplar related of an earlier implementation from De Groote et al., 2015?

**Response #4**

We will add sentences to explain the relationship with ORCHIDEE-SRC by *De Groote et al.* (*2015*) (see **Response #2**).

**Comment #5**

page 4, line 26: "The non-harvested biomass goes to litter"-Should that be really the case? In reality you wouldn't plough or something like that to destroy roots respectively non-harvested biomass. Furthermore you would preserve root mass for a faster growth. Especially for woody plantation, growing out of the stump is a coppice management.

**Response #5**

We agree that it is a rather simple approach to representing the fate of non-harvested biomass in the model for the moment. As the reviewer pointed out, in reality, the root biomass of short rotation coppice poplar and willow will be preserved for growing in the next rotation. Similarly, the root of perennial grasses like *Miscanthus* will also be left for next-year growth. However, the simplistic representation of roots in land surface model (*Warren et al., 2015*), including ORCHIDEE, in particular issues with lacking root phenology and lacking nutrient cycles, calls for introduction of fundamental root processes first. The factors listed by the referee should definitely be considered in the next stage of development to represent the carbon cycle more accurately. However, as we stated in the title and in the introduction, we only aimed to model bioenergy crop yield in this paper. We will thus add sentences in Discussion to incorporate these points on **P15L2**: "In addition to the yields from aboveground biomass, the allocation of belowground biomass also needs to be modified, and the resulting soil carbon stocks need to be evaluated. In the current version, the non-harvested parts of biomass go to the litter pool after each harvest. In reality, however, stumps and coarse roots remain alive in coppicing practices of tree species like eucalypt, poplar and willow, and new shoots grow out of these stumps in the next growing season. Similarly, new shoots grow out of rhizome for perennial grasses like *Miscanthus* in the next growing season after harvest. Carbon in such live biomass compartments does not transfer to the litter or soil and thus does not contribute to soil carbon stocks. It is necessary to correct the model processes in this respect before applying this model to account for the full carbon cycle involving bioenergy plants. Meanwhile, a global observation dataset of belowground biomass and soil organic carbon for bioenergy crops would be desirable to systematically evaluate the model, but does not exists, to the best of our knowledge."

**Reference**

*Warren, J. M., Hanson, P. J., Iversen, C. M., Kumar, J., Walker, A. P. and Wullschleger, S. D.: Root structural and functional dynamics in terrestrial biosphere models - evaluation and recommendations, New Phytol., 205(1), 59–78, doi:10.1111/nph.13034, 2015.*

**Comment #6**

page 5, line 15: Doesn't you need the procedure again for the new implemented PFTs?

**Response #6**

We agree that it will be more precise to use the temperature acclimation parameters explicitly for the specific PFTs like poplar, willow and eucalypt, but these plant types are not included in the 36 plant species in Kattge and Knorr (*2007*). Therefore, we would like to keep the parameters for general PFTs and to be compatible with PFTs other than bioenergy crops.

**Comment #7**

page 6. line 3: It is not clear to me how the parameters are adjusted and how have you evaluated the adjusted parameters?

**Response #7**

We explained how these parameters are adjusted in the following paragraph on **P6L16**: "Specifically for bioenergy crop PFTs, we increased $\theta$ to 0.8 for PFT14 (eucalypt) based on Yin and Struik (2017) and to 0.84 for PFT16 (Miscanthus) based on field measurements from Dohleman and Long (2009). Light use efficiency and productivity are high for bioenergy crops (e.g. see reviews by Forrester, 2013; Heilman et al., 1996; Karp and Shield, 2008; Laurent et al., 2015; Lewandowski et al., 2003; McCalmont et al., 2017; Whitehead and Beadle, 2004; Zub and Brancourt-Hulmel, 2010), and we thus set $\alpha(LL)$ and $g_0$ to the maximum boundary in their ranges from Yin and Struik (2009) to favors high light use efficiency and productivity characteristic of bioenergy cultivars (Table 2)."

Please also see **Response #3** for how we adjusted and evaluated parameters in general.

**Comment #8**

Equation 3: Is Jmax = Jmax25? If not, for which equation you need Jmax? Or please do not confuse the reader by defining Jmax25.

**Response #8**

As shown on **P5L21**, $J_{max25}$ is $J_{max}$ at 25 °C. $J_{max}$ is calculated from $V_{cmax}$ and $r_{JV}$, and $r_{JV}$ is a function of growing temperature. We explained it in equations (1) and (2) on **P5L6-19**.

**Comment #9**

page 7, line 28: To allocate only 20 percent to roots seems to me quite small, as the root turnover leads to a higher loss of root biomass. How is root turnover parametrized?

**Response #9**

As shown in **Fig. S1**, the 20% allocation to belowground is for trees after 20 years for default forest PFT and ca. 10 years for bioenergy trees. For young trees, the carbon allocation to root is higher (20%-80%). This is reasonable since that younger roots have higher respiration rates than the older roots (*Bouma et al., 2001; Fukuzawa et al., 2012*).

In ORCHIDEE, trees lose their fine roots as the same rate that they lose their leaves. Leaf senescence caused by meteorological conditions include cold temperatures, water limitation or both. In addition, a fraction of leaves and fine roots is lost every time step as a function of leaf age based on the fact that trees have to renew the inefficient old leaves, especially for evergreen trees. This was reported and validated in detail in Krinner et al. (*2005*). The emerging evidence of decoupled root and leaf phenology (*Warren et al., 2015*) is not yet represented in land surface models.

**References**

Bouma, T. J., Yanai, R. D., Elkin, A. D., Hartmond, U., Flores-Alva, D. E. and Eissenstat, D. M.: *Estimating age-dependent costs and benefits of roots with contrasting life span: comparing apples and oranges*, New Phytol., 150(3), 685–695, doi:10.1046/j.1469-8137.2001.00128.x, 2001.

Fukuzawa, K., Dannoura, M. and Shibata, H.: *Fine root dynamics and root respiration, in Measuring roots*, pp. 291–302, Springer., 2012.

Krinner, G., Viovy, N., de Noblet-Ducoudré, N., Ogée, J., Polcher, J., Friedlingstein, P., Ciais, P., Sitch, S. and Prentice, I. C.: *A dynamic global vegetation model for studies of the coupled atmosphere-biosphere system*, Global Biogeochem. Cycles, 19(1), doi:10.1029/2003GB002199, 2005.

Warren, J. M., Hanson, P. J., Iversen, C. M., Kumar, J., Walker, A. P. and Wullschleger, S. D.: *Root structural and functional dynamics in terrestrial biosphere models - evaluation and recommendations*, New Phytol., 205(1), 59–78, doi:10.1111/nph.13034, 2015.

**Comment #10**

page 7, line 30: I think not to account for growth out of the stump could cause a deceleration of biomass production which is not realistic, but it also causing to high carbon sequestration into the soil.

**Response #10**

We agree that it is important to account for the stump in the model (see **Response #5**), but we don't fully agree that "it could cause a deceleration of biomass production". We have already evaluated the biomass-age relationship from the model using multiple observation sites (**Section 3.4**). The model generally captured the growth curves from observations (some sites for total biomass of aboveground and belowground, **Figure 4 and 5**), and the model-observation difference can be largely explained by the species varieties and management (see details in **Section 3.4**) that are not explicitly implemented

in the model. Therefore, not accounting for the stump growth does not necessarily lead to a deceleration of biomass production. Otherwise, we may not be able to validate the yields.

We are fully aware that the fate of belowground biomass after harvest is important to derive the full carbon cycle. We have a reserve carbon pool for leaf onset in ORCHIDEE (*Krinner et al., 2005*), and a simple approach to account for the stump is to leave some carbon in this reserve pool after harvest. We will implement this feature and evaluate the belowground and soil carbon in the next step of development, also after collecting new observation data for belowground and soil carbon of different bioenergy crops.

**Comment #11**

page 8, line 21: What is the reason for harvesting in winter at lower biomass harvest? Is that really nutrient recycling? I would assume that you can add nutrients in a managed system.

**Response #11**

Yes, it is due to the nutrient recycling and drying. It could be harvest at maximum yield, and then nutrients need to be added as the reviewer pointed out. However, fertilization increases cost, leaching and $N_2O$ emissions and is neither cost-effective nor environmentally beneficial. In fact, it is recommended to harvest between January and March in the "Planting and Growing Miscanthus – Best Practice Guidelines" by the UK ministry of agriculture (*DEFRA, 2007*).

We will revise the sentence here to make it more clear: "In practice, harvesting of *Miscanthus* and switchgrass is usually performed in winter and early next spring after drying and nutrient recycling through leaf falling off (Lewandowski et al., 2003; Zub and Brancourt-Hulmel, 2010) which leads to a lower biomass at harvest but enhances nutrient conservation. For example, 18%-46% of the nitrogen in *Miscanthus* can be recycled through leaf falling to soil and translocation from shoots to rhizomes (Cadoux et al., 2012). Similar seasonal nitrogen dynamics were also observed for switchgrass (Heaton et al., 2009). In fact, *Miscanthus* is recommended to be harvested between January and March in practice guidelines (DEFRA, 2007). Otherwise, fertilizers have to be applied to amend the nutrient removal from harvest, which is neither cost-effective nor environment-friendly."

**References**

Cadoux, S., Riche, A. B., Yates, N. E. and Machet, J.-M.: Nutrient requirements of Miscanthus x giganteus: conclusions from a review of published studies, Biomass and Bioenergy, 38, 14–22, 2012.
DEFRA: Planting and Growing Miscanthus., 2007.
Heaton, E. A., Dohleman, F. G. and Long, S. P.: Seasonal nitrogen dynamics of Miscanthus × giganteus and Panicum virgatum, GCB Bioenergy, 1(4), 297–307, doi:10.1111/j.1757-1707.2009.01022.x, 2009.
Lewandowski, I., Scurlock, J. M. O., Lindvall, E. and Christou, M.: The development and current status of perennial rhizomatous grasses as energy crops in the US and Europe, Biomass and bioenergy, 25(4), 335–361, 2003.
Zub, H. W. and Brancourt-Hulmel, M.: Agronomic and physiological performances of different species of Miscanthus, a major energy crop. A review, Agron. Sustain. Dev., 30(2), 201–214, 2010.

**Comment #12**

page 9, line 4: But isn't it less practical to harvest in nearly each age class? The harvester could harm other trees. I would assume that plantations consist of homogeneous age classes and are harvested at a certain age. But maybe I do not understand which practise you assume here.

**Response #12**

Yes, plantations in the model are assumed to be homogeneous cohorts in as different patches of a model grid cell and harvested at certain age of maturity. Here, the "boundary" refers to the threshold of biomass to define age classes or cohorts in the model, not the physical boundary between different patches in reality. To avoid misleading, we will revise this sentence as: "Namely, harvesting starts from the second youngest age class, thus the age in the second youngest forest age cohort should be set up as same as the rotation length."

**Comment #13**

page 9, line 15: Are there "real" plantations" already or are that more experimental sites?

**Response #13**

We will add a sentence to clarify it here: "Most of the measurements (>90%) are based experimental trials, especially for *Miscanthus* and switchgrass."

**Comment #14**

page9, line 23: "Note that this dataset does not distinguish the utilization .." - But that makes a big difference.

**Response #14**

First, this is not a problem for *Miscanthus* and switchgrass because they are both designed for bioenergy purpose in the experimental trials. It may influence the woody crops like poplar, willow and eucalypt, but there are not a great number of studies on woody plantation for bioenergy use. Although some plantations are for timber or pulpwood, they can still provide the specific growth information for this woody crop type. We would think it is justified to use these biomass production observations to evaluate the model, considering maybe more uncertainties induced by species and genotype differences and management practices.

**Comment #15**

page 10, line 21: It might be better to count the harvest events.

**Response #15**

Because we artificially harvest 1% of the grid cell each year and re-plant immediately, after the first 5 years (the rotation length), there is always a fraction that is ready for harvest. For example, regrowth of $1^{st}$ year harvest patches will reach the rotation length in the $5^{th}$ year, and the $2^{nd}$ year harvest patches will reach a full rotation in the $6^{th}$ year… Therefore, we used the last 10 years harvested biomass, representing 10 harvested events but probably from different patches.

We will revise the sentence here to clarify it: "The harvested biomass for the last 10 years was used to calculate the median and range of the simulated yields. Note that we artificially harvest 1% of the grid cells each year, and the harvested patches will be planted immediately. After the first 5 years (one rotation length), there is always a fraction reaching a full rotation and ready for harvest. The harvest in the last 10 years thus represents 10 harvest events."

**Comment #16**

page 10, line 25: But this is of enormous importance if you like to estimate biomass potentials for BECCS. It is essential to balance the harvest and the soil carbon losses and the carbon needed for the establishment of a biomass plantation.

**Response #16**

Yes, we agree that soil carbon should be evaluated before using this model to study the full carbon cycle for BECCS. The biomass productivity is relatively isolated from other carbon pools like soil carbon in the model, so the implementation and parameterizations in this study are sufficient to simulate the biomass yields only. The soil carbon evaluation will be conducted in the next step, but it will take time and efforts to collect soil carbon data from observations for different crops. As we stated in the title and introduction, we only aimed to capture the biomass yield observations on global scale in this paper.

**Comment #17**

page 13, line 19: But it seems also that the model underestimates yields in dry regions. Blue rectangle tend to be more left sided for PFT15, 16, and 17.

**Response #17**

As suggested, we will add sentences here to point out this finding: "The strong underestimation (darker blue color) seems more aligned to the drier regions, especially for poplar and willow (PFT15, Fig. 10b)."

**Comment #18**

page 14, line 14: "... different carbon dynamics in litter and soil and water and energy balance can be expected." That's why you need to take for the soil carbon balance as well. This is one of the main issue I have on that manuscript.

**Response #18**

Please see **Response #5**, **#10** and **#16**.

**Comment #19**

page 15, line 2: "... global dataset of soil organic carbon for bioenergy crops to our knowledge." At least you should try to represent the carbon cycle right.

**Response #19**

Please see **Response #5**, **#10** and **#16**.

---

## Author Comment (AC2) · 11 May 2018

**Reviewer #2:**

**Comment #1**

Overall, this manuscript is a straightforward evaluation of a PFT parameterization in a well-established global biogeochemical model. The authors are adding parameterization of specific plants that are used in lignocellulosic biomass for biofuels. The study is motivated by need to connect a global land biogeochemical model, which typically do not have specific parameterization of biofuel crops, to Integrative Assessment Models that include extensive uses of biofuels in many scenarios for energy development.

I appreciate the authors documenting this model developing through a relatively short publication and that the parameters presented are commonly used across other global biogeochemical models. This will allow the manuscript serve as a resource for other modeling groups that add these bioenergy crops to their simulations.

**Response #1**

We thank the reviewer for the comments and suggestions. Please see the detailed point-by-point responses below.

**Comment #2**

My main critique of the manuscript is that it needs more analysis and discussion of causes of the model-data mismatch, specifically the role of management in the parameterization and the observation datasets.

The authors mention that there is considerable variation many of the parameters (e.g., Page 5, line 24). Is that variation related to management? Could there be a parameterization for high intensity management (nutrient additions, irrigation, advanced genetics) and a parameterization for lower intensity management? In general, it would be useful to provide more information about the drivers of variation in the parameters for each species.

**Response #2**

As suggested, we will add sentences to discuss the variations of parameters related to managements on **P14L26**: "
[revised manuscript text omitted]

**Comment #3**

The manuscript focuses on a global analysis, rather than comparing directly to individual field studies. By averaging the studies within a grid-cell, there is considerable variation in the observations within a grid-cell (Figure 3). I assume that much of this variation can be attributed to differences in management of the bioenergy crop. For example, there are likely different levels of nutrient fertilization, irrigation, and use of specific genotypes within a grid-cell. I recommend exploring this variation more. Do the simulations compare better to yields from specific types of management? Addressing this question will help set a path for future model development that includes management practices. For example, if the simulations compare better to the nutrient fertilization treatment trials, then including nutrient limitation will potentially help improve the simulations of the biofuel. I realize that the paragraph on page 11, line 9 address this issue but I found paragraph to be weak. Can the studies not be roughly categorized by management intensity? Furthermore, the final sentence "implying the model is able to capture at least some of the observations in these grid cells" does not give much confidence that the new parameterization is actually an improvement.

**Response #3**

As suggested, we further categorized the observations with different managements (i.e. fertilization, irrigation and species) and added three figures and two tables (reproduced below) to show the model-observation comparison. We also fully discussed the management effects on biomass yields for each bioenergy crop based on evidence from reviews or meta-analyses (*Heaton et al., 2004; Cadoux et al., 2012; Kauter et al., 2003; De Moraes Gonçalves et al., 2004; Wang et al., 2010; Fabio et al., 2018*. See details below). We will also add sentences in the revised manuscript to incorporate these aspects:

"Management like fertilization, irrigation and species plays an important role in the biomass yields. In ORCHIDEE-MICT-BIOENERGY, nutrient limitations and management by irrigation and fertilization are not explicitly implemented. Instead, we used parameter values in the range that favors a higher productivity (Section 2.3, Fig. 1) and compared the simulated yields with the median values of all observations regardless the management (Fig. 3). We further categorized the observations into three groups (fertilization, non-fertilization or non-reported) and compared with simulated yields (Fig. S5). There is no systematic bias between simulated yields and yields at fertilized sites for all PFTs (orange dots in Fig. S5). The model seems to overestimate the yields of eucalypt at sites with non-reported information of fertilization (most gray dots above 1:1 line in Fig. S5a, Table S4) and overestimate yields of poplar and willow at sites without fertilization (green dots in Fig. S5b, Table S4). Yields at sites with non-reported fertilization information are underestimated by the model for *Miscanthus* (gray dots in Fig. S5c, Table S4) but overestimated for switchgrass (gray dots in Fig. S5d, Table S4).

We didn't group the observations based on different fertilization rates because there are large variations in the biomass response to fertilization rates. For example, in a quantitative review by Heaton et al. (2004), the relationship between yields of *Miscanthus* and nitrogen application rates were not significant. Cadoux et al. (2012) reviewed 11 studies that measured *Miscanthus* yields under fertilization, and the biomass response to nitrogen fertilization was positive in 6 of the studies but no response in the others. Similarly, some studies showed positive biomass response of poplar to nitrogen fertilization, but others didn't (Kauter et al., 2003). Eucalypt also showed variable response to fertilization while the general response was positive (De Moraes Gonçalves et al., 2004). In quantitative reviews of fertilization effects on yields of switchgrass (Wang et al., 2010) and willow (Fabio et al., 2018), the relationship between biomass yields and nitrogen fertilization rates was significantly positive but the coefficient of determination ($r^2$) was very low. In summary, biomass response to fertilization varied largely, and evidence from field measurements is not conclusive. More importantly, the basic soil characteristics should be taken into account in addition to the fertilization rates but unfortunately,

we didn't have information of soil nutrient contents nor types, nutrient stoichiometry, rates and timing of applied fertilizers for each site from observations.

We also separated the observations based on irrigation information (irrigation, non-irrigation and non-reported) in comparison with modeled yields (Fig. S6). Both underestimation and overestimation were found for sites with different irrigation management for different PFTs. The yields of eucalypt were underestimated at sites with irrigation (blue dots in Fig. S6a, Table S4) but overestimated at sites with non-reported irrigation information (gray dots in Fig. S6a, Table S4). Compared to fertilization, not many sites reported irrigation information and the quantification of irrigation rates is more difficult. For example, some studies reported irrigation amount per year while some others only reported descriptive information like "soil moisture maintained to field capacity" or "irregular irrigation when necessary".

Comparison between simulated yields and observations for the main species of bioenergy crops is shown in Fig. S7. The model overestimated yields of *Eucalyptus urophylla × E. grandis*, *E. globulus* and *E. nitens* (Fig. S7a, Table S5). For poplar and willow, the model generally overestimated yields of *Populus deltoides × P. nigra*, *P. deltoides* but underestimated yields of *P. trichocarpa* and *Salix schwerinii × S. viminalis* (Fig. S7b, Table S5). There is underestimation of yields for *Miscanthus × giganteus* but overestimation for *Miscanthus sinensis*. In fact, the observed yields of the former are significantly higher than yields of the latter (t-test, p<0.01). Only four sites reported yields for *Panicum pretense*, and they were overestimated by the model (Fig. S7d, Table S5).
"

We also revised the final sentence as: "In addition, the error bars for most sites (67%, 73%, 74% and 64% for PFT14 to PFT17 respectively) reach the 1:1 line (Fig. 3 left panel), implying that at least some observations in these grid cells can be represented by the model.". Here we only stated that although the medians are not on the 1:1 line, some observations can be captured by the model. We didn't imply the improvement after new parameterizations here, because the improvement from the previous model version be clearly seen from **Fig. 11** and discussed in **Section 4.2**.

Fig. S5 Comparison of biomass yields simulated by ORCHIDEE-MICT-BIOENERGY and observations with or without fertilization. Orange, green and gray colors represent the median values of observations with fertilization, without fertilization or non-reported information, respectively in each grid cell. The red line indicates the 1:1 ratio line.

[Figure]

Fig. S6 Comparison of biomass yields simulated by ORCHIDEE-MICT-BIOENERGY and observations with or without irrigation. Blue, green and gray colors represent the median values of observations with irrigation, without irrigation or non-reported information, respectively in each grid cell. The red line indicates the 1:1 ratio line.

[Figure]

Fig. S7 Comparison of biomass yields simulated by ORCHIDEE-MICT-BIOENERGY and observations for the main species of bioenergy crops. Different colors represent the median values of observations for different species in each grid cell. The red line indicates the 1:1 ratio line.

[Figure]

Table S4 Median and 1st and 3rd quartiles of biomass yields under different management practices from observations and the model simulation. N is number of half-degree grid cells with observations.

| PFT | | | 14, eucalypt | | | 15, poplar & willow | | | 16, *Miscanthus* | | | 17, switchgrass | | |
|---|---|---|---|---|---|---|---|---|---|---|---|---|---|---|
| | | | median | 1st quartile | 3rd quartile | median | 1st quartile | 3rd quartile | median | 1st quartile | 3rd quartile | median | 1st quartile | 3rd quartile |
| Fertilization | yes | N | 32 | | | 51 | | | 50 | | | 38 | | |
| | | observation | 18.6 | 13.6 | 24.4 | 9.2 | 7.1 | 11.2 | 12.6 | 9.0 | 16.8 | 9.0 | 5.8 | 10.8 |
| | | model | 17.6 | 15.6 | 18.8 | 9.0 | 6.6 | 10.2 | 11.6 | 9.6 | 14.4 | 9.1 | 8.0 | 9.9 |
| | no | N | 11 | | | 25 | | | 32 | | | 17 | | |
| | | observation | 13.9 | 12.4 | 19.4 | 6.3 | 4.7 | 9.5 | 14.7 | 6.5 | 17.7 | 8.2 | 5.0 | 10.8 |
| | | model | 18.1 | 15.6 | 18.4 | 9.2 | 7.1 | 9.9 | 12.2 | 10.4 | 19.3 | 8.6 | 7.3 | 10.3 |
| | non-reported | N | 28 | | | 57 | | | 21 | | | 8 | | |
| | | observation | 11.9 | 10.1 | 16.3 | 7.1 | 5.1 | 9.8 | 15.0 | 12.4 | 19.0 | 8.5 | 5.6 | 9.1 |
| | | model | 17.8 | 15.1 | 19.3 | 7.1 | 5.5 | 8.9 | 9.3 | 8.6 | 11.3 | 9.9 | 9.1 | 10.7 |
| Irrigation | yes | N | 13 | | | 19 | | | 12 | | | 0 | | |
| | | observation | 25.4 | 17.3 | 26.4 | 8.6 | 6.0 | 10.2 | 14.2 | 8.2 | 19.7 | | | |
| | | model | 17.1 | 14.2 | 19.5 | 8.9 | 7.0 | 10.0 | 9.5 | 8.1 | 15.0 | | | |
| | no | N | 13 | | | 15 | | | 14 | | | 2 | | |
| | | observation | 18.3 | 13.2 | 22.4 | 7.6 | 5.4 | 9.4 | 8.5 | 4.1 | 16.7 | 8.0 | 7.4 | 8.5 |
| | | model | 18.2 | 15.2 | 19.5 | 9.1 | 6.4 | 10.1 | 9.4 | 8.7 | 11.0 | 5.4 | 4.1 | 6.7 |
| | non-reported | N | 45 | 0 | 0 | 95 | 0 | 0 | 51 | 0 | 0 | 41 | 0 | 0 |
| | | observation | 14.7 | 11.0 | 21.0 | 8.0 | 5.8 | 10.0 | 13.8 | 10.2 | 15.3 | 8.1 | 5.7 | 10.0 |
| | | model | 17.6 | 15.2 | 19.3 | 8.5 | 6.2 | 10.0 | 11.3 | 9.7 | 13.8 | 9.1 | 7.7 | 9.9 |

Table S5 Median and interquartiles of biomass yields for the main species from observations and the model simulation. N is number of half-degree grid cells with observations.

| | N | observation | | | model | | |
|---|---|---|---|---|---|---|---|
| | | median | 1st quartile | 3rd quartile | median | 1st quartile | 3rd quartile |
| *Eucalyptus urophylla x Eucalyptus grandis* | 7 | 17.7 | 14.9 | 20.2 | 18.4 | 17.9 | 20.7 |
| *Eucalyptus grandis* | 12 | 17.8 | 15.2 | 21.3 | 18.8 | 14.7 | 22.5 |
| *Eucalyptus globulus* | 12 | 10.8 | 9.5 | 13.8 | 15.7 | 12.9 | 17.2 |
| *Eucalyptus nitens* | 2 | 7.9 | 6.0 | 9.7 | 18.9 | 18.5 | 19.3 |
| *Populus tristis* | 2 | 6.7 | 6.3 | 7.0 | 7.3 | 7.2 | 7.5 |
| *Populus deltoides x Populus nigra* | 13 | 6.8 | 4.9 | 7.7 | 9.9 | 8.9 | 11.0 |
| *Populus trichocarpa x Populus deltoides* | 7 | 11.4 | 5.4 | 16.2 | 8.7 | 6.9 | 10.0 |
| *Populus trichocarpa* | 19 | 9.8 | 6.8 | 11.8 | 7.9 | 6.6 | 10.1 |
| *Populus deltoides* | 14 | 7.5 | 5.7 | 13.5 | 9.5 | 5.5 | 12.9 |
| *Salix viminalis* | 17 | 8.9 | 7.7 | 10.0 | 8.3 | 5.9 | 9.1 |
| *Salix schwerinii x Salix viminalis* | 7 | 11.6 | 10.3 | 12.3 | 8.3 | 5.8 | 8.8 |
| *Salix viminalis x Salix viminalis* | 4 | 8.8 | 7.7 | 10.0 | 8.4 | 7.7 | 8.7 |
| *Miscanthus x giganteus* | 51 | 14.6 | 10.1 | 18.8 | 10.7 | 8.7 | 13.8 |
| *Miscanthus sinensis* | 22 | 8.6 | 4.8 | 12.2 | 10.6 | 9.5 | 13.0 |
| *Panicum virgatum* | 39 | 8.9 | 6.1 | 10.4 | 9.1 | 7.7 | 9.9 |
| *Panicum pratense* | 4 | 3.5 | 3.5 | 4.5 | 7.9 | 7.2 | 8.2 |

**Comment #4**

Also, these is an issue for the editors to provide input on, but the paper leans heavily on a data paper that is submitted to another unnamed journal. Therefore, a reviewer of this paper is unable to comment on the quality and applicability of the observational dataset. Should this paper be allowed to be published before that data paper is available?

**Response #4**

As shown on **P9L15-24**, we briefly reported information on the dataset related to this study. We already submitted the revised version of the dataset after peer-review in a data journal. If the dataset paper is accepted before this GMD manuscript, we will provide the detail reference information. The dataset will be eventually available to public and free to access (hopefully soon).

**Comment #5**

The spatial mapping of the model-bias is useful but it opened the question whether there are spatial differences in management that could explain the spatial variation in the mismatch.

**Response #5**

We agree that management would contribute to the spatial mismatch between model and observation. However, it is difficult to isolate individual management factor (e.g., species, irrigation and fertilization) or systematically evaluate the role of all these factors in driving model-observation mismatch. In addition, if we separate the spatial maps of sites with a specific management, the number of sites is limited in most cases and consequently, no spatial patterns can be observed. We thus discussed the management effects on the biases between model and observation globally as suggested by the reviewer (see **Response #3**) but didn't analyze further the regional management contributions to the spatial patterns of mismatch here.

**Specific Comments:**

**Comment #6**

The model evaluation and discussion sections blur together a bit at the edges (section 4.1 seems like a continuation of section 3). I recommend making the separation more clear.

**Response #6**

We will move section 4.1 from **Discussion** to **Model evaluation** section.

**Comment #7**

Section 3.3 says that the model-observations results generally lie around the 1:1 ratio line but doesn't provide any statistics on the fit. What is the slope and intercept from the 1:1 fit?

**Response #7**

We will add sentences here to report some statistics: "Although the regression between modeled and observed medians is not significant with a low $r^2$ value because of the overestimation and underestimation at some sites (Fig. 3 left panel), the difference between the two samples of modelled and observed yields is not significant (t-test, $p>0.17$) and the percent bias (PBIAS, defined as sum of biases divided by sum of observed values, Moriasi et al., 2007) ranges from 2% to 8% for all PFTs,

implying that the global distributions of modeled and observed yields are consistent (Fig. 3 right panel). In addition, the error bars for most sites (67%, 73%, 74% and 64% for PFT14 to PFT17 respectively) reach the 1:1 line (Fig. 3 left panel), implying that at least some observations in these grid cells can be represented by the model."

**Reference**

*Moriasi, D. N., Arnold, J. G., Van Liew, M. W., Bingner, R. L., Harmel, R. D. and Veith, T. L.: Model evaluation guidelines for systematic quantification of accuracy in watershed simulations, Trans. ASABE, 50(3), 885–900, 2007.*

**Comment #8**

Figure 6. It is hard to see the gridcells in the subboxes. For example, box 2 in Figure 6 has lower points that are impossible to see. Can the subboxes be bigger. I also recommend adding a histogram inset that summarizing the data across grid-cells for all the similar figures (Figure 6-9)

**Response #8**

We will enlarge box 2 in Figure 6 and add a histogram inset in Figure 6-9 as suggested.

**Comment #9**

Figure 10 stated that there is a 1:1 line that is not present in the figure

**Response #9**

We will delete this sentence in Figure 10 caption.

**Comment #10**

Page 3 Line 25:Change "ORHCIDEE" to "ORCHIDEE"

**Response #10**

We will revise it accordingly.

**Comment #11**

Page 8, line 22: change 'through leaf falling off' to 'though leaf senescence'

**Response #11**

We will revise it accordingly.

**Comment #12**

Page 9 Line 8: Change "corresponding" to "corresponds".

**Response #12**

We will revise it accordingly.

**Comment #13**

Page 11 Line 27:Change "after plantation" to "after planting"

**Response #13**

We will revise it accordingly.

**Comment #14**

Page 12 Line 13:It is unclear what is meant by "because of the large spacing of plantation the trial experiment which results in . . .". Perhaps what was intended was something like: "because of the large spacing of the planting in the trial at that experimental

site, which results in . . .".

**Response #14**

We will revise it accordingly.

**Comment #15**

Page 13 Line 9: Change "US" to "the US

**Response #15**

We will revise it accordingly.